# Origin of water in the Badain Jaran Desert, China: New insight from isotopes

Xiujie Wu [1, 2, 3], Xu-Sheng Wang[1], Yang Wang [2, 3, 4,*], Bill X. Hu [1, 2, 4,*]

[1]School of Water Resources and Environment, China University of Geosciences (Beijing), Beijing, 100083, P. R. China
[2]Department of Earth, Ocean, and Atmospheric Science, Florida State University, Tallahassee, FL, 32304, USA
[3]High National Magnetic Field Laboratory, Tallahassee, FL, 32310, USA
[4]Institute of Groundwater and Earth Sciences, The Jinan University, Guangzhou, Guangdong Province, 510632, P. R. China
*Correspondence to*: Yang Wang (ywang@magnet.fsu.edu); Bill X. Hu (bill.x.hu@gmail.com)

**Abstract.** To better understand the origin of water in the Badain Jaran Desert, China, water samples were collected from lakes,
a spring and local unconfined aquifer for analyses of radiocarbon ($^{14}$C) and tritium ($^3$H), stable hydrogen and oxygen isotope
ratios ($\delta^2$H - $\delta^{18}$O), and *d*-excess values (=$\delta^2$H – 8$\delta^{18}$O). A series of evaporation experiments were also conducted in the desert
to examine how the isotopic signature of water may change during evaporation and infiltration under local environmental
conditions. The results show that the lakes in the southeastern sand dune area are fed by groundwater discharging into the lakes
and that local groundwater, on the other hand, is derived primarily from modern meteoric precipitation in the region. Although
dissolved inorganic carbon (DIC) in groundwater yielded very old radiocarbon ages, the presence of detectable amounts of
tritium in groundwater samples, together with their $\delta^2$H, $\delta^{18}$O and *d*-excess characteristics, strongly suggest that the old
radiocarbon ages of DIC do not represent the residence time of water in the aquifer but are the result of addition of old DIC
derived from dissolution of ancient carbonates in the aquifer. The data do not support the hypothesis that the water in the
Badain Jaran Desert was sourced in remote mountains on the northern Tibetan Plateau. This study also finds no support for
the hypothesis that present-day water resources in the desert were recharged by the precipitation that fell in the past during the
early Holocene when the climate was much wetter than today. Instead, this study shows that both groundwater and lake water
were originated from meteoric precipitation in the region including mountainous areas adjacent to the desert under the modern
climatic condition.

## 1 Introduction

Arid regions comprise nearly one third of the Earth's land area. In these regions, surface water and groundwater are scarce,
due to the low precipitation and strong evaporation. The scarcity of freshwater poses serious challenges to agriculture and
economic development. Hence, it is critical to identify the origins of surface- and ground-waters in these regions for proper
management and protection of the limited water resources. Although water isotopes have been widely used to identify water
sources (e.g., Rademacher et al., 2002; Gates et al., 2008a; Morrissey et al., 2010; Chen et al., 2012), the isotopic composition
of groundwater and surface water in arid environments can deviate significantly from that of local precipitation due to

evaporation, which complicates the interpretation of water stable isotope data and can lead to equivocal inferences about recharge origin (e.g., Gates et al., 2008a, b; Chen et al., 2012). Previous studies have shown that evaporation results in characteristic water isotopic patterns controlled by local climatic conditions, which could provide useful insight into the mechanisms of recharge (e.g., Clark and Fritz, 1997; Geyh et al., 1998; Oursingbé and Tang, 2010; Tweed et al., 2011; Wood, 2011; Murad and Mirghni, 2012). Despite the progress in the application of isotopes and other geochemical tracers in hydrological studies, the origin and residence time of groundwater in many arid and semi-arid areas remain open questions in water resource research.

The Badain Jaran Desert (BJD) is a hyper-arid area located in the western Inner Mongolia, China. The landscape in the BJD is dominated by numerous large sand dunes, interspersed with more than 100 lakes and springs. Most of the lakes are permanent surface water bodies, with no obvious drying-up trends. The presence of these apparently permanent lakes in this hyper-arid environment has attracted many researchers to the region to investigate the origin of the lake water (Geyh and Gu, 1992; Geyh et al., 1998; Chen et al., 2004; Ma and Edmunds, 2006; Gates et al., 2008a; Yang et al., 2010; Zhang et al., 2011; Zhao et al., 2012; Wu et al., 2014). Because of the harsh environment and the lack of basic meteorological, climatic and geologic data in the desert, previous studies relied primarily on the isotope analyses of water samples collected from limited localities and times to draw inferences about the source and/or flow path of water in the area. It is generally agreed that groundwater is the main water source to the lakes and the groundwater inputs into the lakes balance the water loss due to intense evaporation as continuous groundwater discharge has been spotted around and under the lakes (Chen et al., 2012; Dong et al., 2013). However, the origin of groundwater in the desert remains a hotly debated issue (see review by Dong et al., 2013). Earlier studies suggest that groundwater in the BJD originated directly from precipitation in the area and the numerous large and highly permeable sand dunes serve as an effective storage for groundwater discharging into the lakes (Hoffmann, 1996; Jakel, 2002; and Wang, 1990). Several recent studies, however, suggest that the groundwater in the BJD was originated in neighboring basins or adjacent mountains (Dong et al., 2004; Gates et al., 2008a; Gates et al., 2008b; Ma and Edmunds, 2006; Ma et al., 2007). Other studies suggest that the lake water and groundwater were recharged by local precipitation during the early Holocene when the climate was much wetter than today (Yang and Williams, 2003; Yang, 2006; Yang et al., 2010). It has also been hypothesized that the water in the area may be originated in high mountains in the Qilian Mountatins and was transported over long distances through deep and large fractures (fault system) underground to the BJD (Chen et al., 2004; Ding and Wang, 2007).

In this study, we present new stable and radioactive isotope data from water samples collected from the local aquifers and lakes as well as from evaporation experiments. The new data are used, in conjunction with data published in the literature, to test the various hypotheses regarding the origin of groundwater and to develop a new conceptual model for groundwater recharge-discharge in the dune-lake systems of the BJD.

## 2 Study area

The Badan Jaran Desert is located on the northwestern Alxa Plateau in China, stretching from 39°04′N to 41°12′N,99°23′E to 104°34′E (Fig.1). It is the third largest desert in China and the world's fourth largest desert. It is bordered by mountains (i.e., Zongnai Mountain, Yabulai Mountain, and Beida Mountain) to the east and south, and by the Gurinai grassland and Guazi lacustrine plains to the west and north. The desert covers approximately 50,000 km$^2$. The landscape in the desert consists of numerous unconsolidated mega-dunes (> 300 m in height) dotted with many lakes and springs in the southeastern region of the desert that covers a total area of ~14,000 km$^2$. Most of the lakes are concentrated within an area of about 2500 km$^2$ around the site C in Fig. 1A. The total water surface area of these lakes is ~22.3 km$^2$ (Yang et al., 2010). The lakes and springs are thought to be fed by local groundwater. The lakes have no surface inflows and outflows.

In the southern part of the desert, vegetation coverage is less than 10% and consists of a few grasses and shrubs with long roots horizontally distributed in the sand to maximize the extraction of soil water. Interlayer structures beneath the dune surface consisting of aeolian sand, flood deposits and lacustrine deposits in the lowlands between the mega-dunes were found from sediment cores drilled at several locations (Gao et al., 1995). Extensive calcareous cementation and cemented tubes of dead plant roots were found in the slopes of the dunes and there are tufa deposits at some of these lakes (Yang, 2000). Outcrops of Jurassic, Cretaceous, and Tertiary sandstones can be seen along the fringe of the desert (Ma, 2002). According to Ma et al. (2007) and Wang et al. (2014), groundwater mainly exists in the continuous unconfined aquifer that consists of Quaternary sandy sediments in the center of the BJD. This aquifer varies in thickness from ~ 100 to ~300 m and connects directly to the lakes (which are all less than 20 m deep). The underlying Tertiary clay and fine sandstone formation serves as the uppermost confining bed for the deep confined aquifers that consist primarily of Tertiary and Cretaceous sandstones, with a total thickness exceeding 1000 m. A borehole (80 m deep) near the Badain Lake showed that the Quaternary sands comprise a shallow phreatic aquifer at the edge of the desert but may receive deep groundwater discharge from the Yabulai Mountain (Gates et al., 2008a). In the southeastern part of the BJD, groundwater level decreases from the mountain areas to the north and west, driving a regional flow toward the desert area where lakes are concentrated in.

The southeastern region of the Badain Jaran Desert is near the present-day northern boundary of the East Asian summer monsoon regime. Summer monsoons provide the primary source of precipitation, which accounts for ~ 70% of the annual precipitation in this region (Gao et al., 2006). Cold and dry continental air masses from prevailing westerly winds dominate in winter (Gates et al., 2008a). The area has an extremely arid temperate continental climate, characterized by hot summers and cold and dry winters. Diurnal variation of the temperature is high, which is about 10~30°C, with high temperatures ranging up to 41°C during daytimes in summer months. The mean daytime temperature is 36°C in July and −12°C in January, respectively.

The annual precipitation decreases from 120 mm/a in the southeast to less than 40 mm/a in the northwest region of the Badain Jaran Desert (Domrös and Peng, 2012). Because of the arid climate, the desert has extremely high rates of potential evaporation from water surfaces; estimates ranged from 2600mm/a (Ma et al., 2003) up to 4000mm/a (Chen et al., 2004). However, a recent study (Yang et al., 2010) suggests that the potential evaporation is only 1040 mm/a, much lower than previously thought.

## 3 Methods

### 3.1 Field Sampling

Water samples for this study were collected in August 2012 and July 2013 mainly from the southeastern part of the desert
where lakes are concentrated. Samples were collected from various lakes, including the Sumu Jaran Lake (~0.6 km$^2$), Sumu Barun Jaran Lake (~1.3 km$^2$), Yindeertu Lake (~0.9 km$^2$), East Badain Lake (0.03 km$^2$) and West Badain Lake (0.08 km$^2$), as well as from domestic wells and a spring (Fig. 1). The Sumu Barun Jaran Lake is the second largest saline lake in the BJD whereas the East Badain Lake is one of a few small lakes with relatively fresh lake water. The maximum depths of water in the lakes range between 1 m and 13 m. We collected groundwater samples from four wells around the Sumu Jaran Lake and
Sumu Barun Jaran Lake (Fig. 1C). The wells are shallow (maximum depth is 16 m) and screened with stone blocks on the wall, being utilized every day to supply drinking water for the local people. In addition, two sets of samples were collected from different depths in these two lakes (Fig. 2A). The deep samples from the lakes were collected with a peristaltic pump with a tube penetrating into the lake and another tube discharging water out. The water was pumped through a layer of gauze and a layer of paper filter into 10 mL glass bottles. Before a water sample was collected into a sample bottle, five minutes
pumping was performed to flush the sampling system and each sampling bottle was rinsed three times with the water to be sampled. For groundwater in wells, sampling was carried out after a 20-minutes pumping to flush the sampling system. Groundwater samples were collected in 600 mL Pyrex Brand Square bottles for radiocarbon ($^{14}$C), [HCO$_3^-$], and tritium ($^3$H) analyses. Each bottle was sealed carefully to avoid the existence of bubbles within the water sample.

Each groundwater sample for $^{14}$C analysis was dosed with a drop (~50 $\mu$l) of 20M NaOH to poison the samples to inhibit
microbial activity before the bottle was sealed. The samples were sent to Beta Analytic Radiocarbon Lab in Miami (Florida) for the radiocarbon analysis. Because the desert area is sparsely settled, there are only a few wells from which we could obtain samples of the groundwater. The existing wells are generally less than 20 m deep and the depths to water table are less than 4 m. We have also collected a water sample (SY-1) from a spring near the Yindeertu Lake (Fig. 1C).

There was a precipitation event during the week of our field exploration in the Sumu Jaran Lake area in August 2012. It rained at night and continued into the early morning. A rain water sample was collected from a puddle on the ground. This sample, combined with previously published precipitation isotope data (Gates et al., 2008a) and data from Zhangye - the nearest IAEA-GNIP (International Atomic Energy Agency Global Network of Isotopes in Precipitation) station located approximately 170 km to the southwest of the study area at the altitude of 1400 m, (IAEA/WMO, 1986-2003), were used to approximately
represent the isotopic composition of the local precipitation in the BJD.

## 3.2 Evaporation experiments

To better understand the effects of strong evaporation on the isotopic characteristics of water in the desert, we conducted surface water evaporation experiments and soil water infiltration-evaporation experiments on the shore of the Sumu Jaran Lake (Fig. 2B) in July 2013. The observation site and experimental methods are schematically presented in Fig. 2B&C.

Evaporation from open water was monitored in two experiments using local groundwater (Pan-1) and lake water (Pan-2), respectively (Fig. 2C). The two plastic pans, which have the same cylinder shape with a diameter of 21 cm and height of 10 cm were used in the evaporation experiments. They were filled with water to an initial depth of 6 cm and then put on a flat ground surface to allow water to evaporate. With continuous evaporation, the water levels in the pans gradually decreased. We measured the water depths every day at 7 a.m. and 7 p.m. Isotope samples from both pans were collected at 7 p.m. every day

after the measurements of the water level. These experiments continued for 6 days. The air temperature and humidity at the height of 2 m above the ground surface were also measured every two hours in daytimes. A total of 6 water isotope samples were obtained from each of the experiments. During the period of the experiments, the air temperature varied between 17°C and 42°C and the humidity varied between 20% and 70%.

Evaporation of soil water was observed in three experiments ES-1, ES-2, and ES-3 (Fig. 2C). Three pits were excavated to

different depths below a flat ground surface to install the evaporation-infiltration systems. Each of the evaporation-infiltration systems was composed of a sandbox with a funnel and a bottle attached to the bottom. The sandbox is 0.5 m× 0.5 m wide, with a depth of 9 cm, 15 cm and 22 cm, respectively, for ES-1, ES-2 and ES-3. The sands were initially dry but were wetted slightly with fresh water from a nearby well (WS-4) using a small sprinkler when the sands were put into the box layer by layer. Artificial rainfall with 250 mL volumetric water in 6 minutes was made twice a day during the daytime using a watering

can (the mean "rainfall" intensity is 0.167 mm/min) for each sandbox to increase the soil moisture content. After the artificial rainfall, some of the soil water was lost due to evaporation and some became infiltration water that passed through the sandbox and was collected in the bottle below the sandbox. After 6 days, the bottles were dug out and the water samples from the bottles were immediately transferred into sampling vials for isotope analysis later in the lab. We obtained 3 samples of infiltration water (ES-1, ES-2 and ES-3) for isotope analysis.

## 3.3 Laboratory methods

The water samples were collected for various chemical analyses. Among them, 21 were prepared for stable hydrogen and oxygen isotope analyses, including 12 lake water samples, 1 spring sample, 1 precipitation sample and 7 groundwater samples. The depths of water tables and the conductivities of water bodies in the desert were measured using a TLC (temperature, level, conductivity) meter (Model 107 TLC, Solinst). Seven groundwater samples were analyzed for radiocarbon contents of

dissolved inorganic carbonate (DIC) and four samples were analyzed for tritium ($^3H$).

The radiocarbon contents of DIC samples were analyzed using an accelerator mass spectrometer. The $^{13}C/^{12}C$ ratio was also measured relative to the VPDB (Vienna Pee Dee Belemnite) standard for the radiocarbon age correction. Radiocarbon dating

was performed by Beta Analytic, Inc. (Miami, Florida, USA). The radiocarbon age represents the measured radiocarbon age corrected for isotopic fractionation using the $\delta^{13}C$. The $HCO_3^-$ and Tritium contents of the groundwater samples were measured using Metrohm automatic titration apparatus and a low background liquid scintillation spectrometer (Quantulus 1220-003), in the Analytical Laboratory of Beijing Research Institute Of Uranium Geology. The analytical uncertainty of the Tritium analysis has a level from ±0.4 TU to ±0.7 TU. Stable isotope samples were analyzed using a Finnigan Gas Bench II Auto Carbonate/Water Device interfaced to a Finnigan MAT delta PLUS XP stable isotope ratio mass spectrometer in the Stable Isotope Laboratory at Florida State University. The stable isotopic results are reported in the standard notation as δD, $\delta^{18}O$ and $\delta^{13}C$ values Eq. (1):

$$\delta = \left(\frac{R_{SA}}{R_{STD}} - 1\right) \times 1000‰, \tag{1}$$

where $\delta$ is the isotopic concentration of a sample, $R_{SA}$ is the isotope atom ratio D/H, $^{18}O/^{16}O$ or $^{13}C/^{12}C$, $R_{STD}$ is the corresponding isotope atom ratio of the international standard V-SMOW for hydrogen and oxygen and V-PDB for carbon. The $\delta^2H$ and $\delta^{18}O$ values are normalized to the VSMOW-SLAP scale (Vienna Standard Mean Ocean Water), with a reproducibility (±1σ) of about ±1‰ and ±0.1‰, respectively. The precision for DIC- $\delta^{13}C$ is ±0.3‰ (±1σ).

## 4 Results

### 4.1 Evaporation experiments

δD and $\delta^{18}O$ values of the water samples collected from the pan-evaporation experiments and evaporation-infiltration experiments are shown in Table 1 and Figure 3. The residual volume of water in the pans after 5 days of evaporation decreased to less than 50% of the initial volume while the water isotope contents became progressively enriched along an evaporation line $EL_1$ (Fig. 3a). The local evaporation line ($EL_1$) established through our pan evaporation experiments has a slope of 4.6 (Fig. 3a). As a result of evaporation, the $\delta^{18}O$ (δD) of water in Pan-1 increased from −3.5‰ to 23.2‰ (−46‰ to 66‰ for δD), and the $\delta^{18}O$ (δD) of water in Pan-2 increased from 7.6‰ to 18.2‰ (10‰ to 70‰ for δD). The isotopic enrichment factor is ∼ 0.2‰ for $\delta^{18}O$ and 1‰ for δD per 1% water loss, respectively. The d-excess values (=$\delta^2H – 8\delta^{18}O$) of remaining water, on the other hand, decreased with the evaporation (Fig. 3b) from −18‰ to −120‰ in Pan-1 and from −51‰ to −76‰ in Pan-2. Note that Pan-1 and Pan-2 were initially filled with groundwater and lake water, respectively. The EC results shown in Table 2 indicate fresh groundwater (EC<1 mS/cm) and saline lake water (EC is generally higher than 100 mS/cm). Similarly, heavy isotopes in the water samples collected from the evaporation-infiltration experiments (i.e., ES-1, ES-2 and ES-3) were progressively enriched with evaporation (Fig. 3). The d-excess values of infiltrated water (minimum −35‰) were also significantly lower than that of the artificial rainfall water (−18‰).

*Insert Table 1*

*Insert Figure 3*

## 4.2 Isotope compositions of natural waters

The $\delta^2H$ and $\delta^{18}O$ values of natural water samples are shown in Table 2. Water samples collected from different depths in both of the Sumu Jaran Lake (LX-1 to LX-4) and the Sumu Barun Jaran Lake (SX-1 to SX-4) exhibited little isotopic variations with depth. For Sumu Jaran Lake, the $\delta^{18}O$ of water collected from 0, 1.5 m, 3 m and 5 m depths varied from 7.0‰ to 7.4‰,

with a mean of 7.3 ±0.2‰ (1 standard deviation from the mean) while the δD ranged from 1‰ to 5‰, averaging 2 ±2 ‰. For the Sumu Barun Jaran Lake, the average $\delta^{18}O$ and δD values of water samples from 0, 2 m, 4 m and 6 m depths are 5.9 ±0.1‰ (ranging from 5.7‰ to 6.0‰) and −1 ±1‰ (ranging from −3‰ to 1‰), respectively. The relatively small variation of $\delta^{18}O$ and δD with depth for such groundwater-fed lakes suggests that the lake water was well mixed, which is confirmed by the measurements of temperature and electric conductivity profiles in the summer (Chen et al., 2015). Similar results were reported

by Wu et al. (2014) for the Nuoertu Lake in the same desert.

*Insert Table 2*

The radiocarbon activities in groundwater samples, reported as pMC (percent Modern Carbon) range from 35.81 to 72.71, and the $\delta^{13}C$ values vary from −11.1‰ to −6.7‰ (Table 3). The lowest and highest radiocarbon activities are found in the samples from wells near the Sumu Jaran Lake (WS-1 and WS-3). The tritium activities in the groundwater samples range from 0.4 TU

(WS-1) to 5.4 TU (WS-3) and are positively correlated with the radiocarbon activities. The tritium levels in our samples are significantly less than that of the single groundwater sample W34 (43.2 TU) collected in 2004 from the Yabulai Mountain (Fig. 1A) reported in Gates et al. (2008).

*Insert Table 3*

## 5 Discussions

### 5.1 The evaporation effects

It is well known that evaporation preferentially removes the light isotopes from the liquid phase, resulting in a progressive isotopic enrichment in the remaining water following an evaporation line in a $\delta^{18}O$ *vs.* δD plot (e.g., Gonfiantini, 1986). The slope of the evaporation line varies from 2 to 5 depending on local climatic conditions. The evaporation line in the study area established through our evaporation experiments has a slope of 4.6 (i.e., EL$_1$ in Fig. 3a). The $\delta^{18}O$ and δD values of the

infiltration water samples collected from the infiltration-evaporation experiments also fall on the evaporation line established through pan evaporation experiments (Fig. 3a). This agreement indicates that evaporation during infiltration occurs and imparts a characteristic isotopic trend reflecting local climatic conditions. These experiments demonstrate that in the present-day arid environment infiltration water could undergo significant isotope enrichment due to evaporation as rainwater descends through the atmosphere and infiltrates into the ground to recharge the groundwater. The similarity of the evaporation effects between

the pan water and infiltration water samples may be caused by two reasons. First, surface evaporation is the dominant evaporation process in the sand boxes without significant influences from pore vapour transport and vegetation uptake. Second,

the stable isotopes fractionation in evaporation processes within the sands is similar to that in the surface evaporation. Laboratory experiments also show that the isotope fractionation of water evaporation in unsaturated sands is similar to surface water evaporation (Sun et al., 2009). However, we had only three samples in the *in situ* infiltration-evaporation experiments, which may be not enough to represent the general behaviours of water in the desert, more experimental studies are required to confirm this finding. For groundwater recharged by direct filtration through the unsaturated zone in arid regions, isotopic enrichment due to evaporation during infiltration need to be considered when interpreting water isotope data. Our evaporation experiments also show that *d*-excess values of water decrease progressively with increasing evaporation and were negatively correlated with the $\delta^{18}O$ values (Fig. 3b). Although the *d*-excess values are often used to infer atmospheric vapor sources, the evaporation experiments show that the *d*-excess values of water in the study area are primarily controlled by evaporation and decrease significantly but systematically with the extent of evaporation, providing another fingerprint for tracing the locally recharged water. Linear regression of the experimental data yielded an evaporation trend line with a slope of −3.4 in the *d*-excess *vs.* $\delta^{18}O$ plot for the study area (Fig. 3b).

The evaporation experiments were conducted in the summer and may not represent evaporation conditions in other seasons. However, the similarity between the evaporation line determined through our evaporation experiments (Fig. 3a) and those derived from measurements of natural water samples (Fig. 4) implies that seasonal variations in meteorological conditions do not significantly alter the evaporative $\delta D$-$\delta^{18}O$ pattern in the desert.

## 5.2 $\delta^{18}$O, $\delta^{2}$H and *d*-excess of water in the BJD and surrounding areas

Water samples collected from various lakes in the BJD are significantly enriched in the heavy isotopes $^{18}O$ and D relative to the groundwater samples (Fig. 4). For example, the mean $\delta D$ value of the samples in lakes is about 46‰ higher than that of the groundwater samples. However, all the $\delta^{18}O$ and $\delta D$ values of groundwater and lake water samples fall below the Global Meteoric Water Line (GMWL) along with a trend line (Fig. 4), indicating that they have been affected by evaporation (Gonfiantini, 1986). This trend line (i.e., $EL_2$ in Fig. 4) is nearly identical to the local evaporation line $EL_1$ (Fig. 3a) established through the evaporation experiments, indicating that lakes and groundwater in the study area have evolved from meteoric precipitation under modern or similar to modern climatic conditions. This rules out the hypothesis that the lakes were recharged by groundwater that derived from precipitation during the early Holocene when the climate was much wetter than today (Yang and Williams, 2003; Yang et al., 2010). This also confirms that the lake water is originated from groundwater but has evolved into a more $^{18}O$/D-enriched state due to higher degrees of evaporation.

Previous studies have also determined evaporation line in the BJD (Chen et al., 2004, 2012; Wu et al., 2014). The evaporation line reported by Wu et al. (2014) (i.e., $\delta D = 4.2\ \delta^{18}O − 31.4$; sample number n = 110; trend-line fit $R^2 = 0.92$) is very similar to the $EL_2$ (Fig. 4). Although Chen et al. (2004) reported an EL that had a slope of 5, higher than that of the $EL_2$ ($\delta D = 4.5\ \delta^{18}O – 27.7$), they did not provide sufficient information about their samples and $\delta D$-$\delta^{18}O$ results to allow for comparison with our results. However, their later study is more informative as it reported the sampling sites and the $\delta D$-$\delta^{18}O$ results (Chen et al., 2012), which showed similar results to ours in the hinterland area of the BJD where our data were collected. It appears that

the higher slope and lower intercept can be attributed to the samples that were collected from areas located on the northern and western margins of the BJD, Wentugaole and Gurinai (Chen et al., 2012). These two sites, especially the Wentugaole, is too far away from our sampling sites (Fig. 1A).

The intersection of the evaporation line and the GMWL in the $\delta^{18}$O-$\delta$D plot suggests that the meteoric precipitation recharging the local groundwater system had average $\delta^{18}$O and $\delta$D values of −10.8‰ and −76‰, respectively (Fig. 4). They are significantly lower than those of the sole rain sample that we collected in the BJD in August of 2012 (Fig. 4). This does not necessarily mean that groundwater and lake water in the BJD were not recharged or affected by local precipitation because the number of precipitation samples is too small to capture the full range of isotopic variation of local precipitation. The precipitation isotope data for the period of 1986-2003 from the nearby IAEA-GNIP station in Zhangye show that the isotope composition of precipitation in the area could vary considerably due to variations in atmospheric conditions, with summer precipitation having higher $\delta$D and $\delta^{18}$O values than winter precipitation (Fig. 4). The single rainwater sample that we collected in the summer yielded $\delta$D and $\delta^{18}$O values similar to those of summer precipitation measured at the IAEA station in Zhangye (Fig. 4). The weighted mean $\delta^{18}$O and $\delta$D values of the annual precipitation (1986-2003; 70 samples) in Zhangye are also very close to the inferred $\delta^{18}$O and $\delta$D values of meteoric precipitation feeding the shallow aquifer in the BJD (Fig 4). This is consistent with the view that the groundwater in the BJD was originated from modern precipitation in the region (Hoffmann, 1996; Jakel, 2002).

*Insert Figure 4*

Although a number of studies have used the natural abundances of stable isotopes to infer the origin of water in the BJD (Chen et al., 2004; Ding and Wang, 2007; Gates et al., 2008a; Gates et al., 2008b; Ma and Edmunds, 2006), the effects of evaporation on the isotopic ratios and the *d*-excess value of water were not systematically analyzed. Here, we compare our new isotope data with previously published data (Gates et al., 2008a; Zhang et al., 2011; Zhao et al., 2012) and discuss their implications for the origin of water in the BJD.

As shown in Fig. 5a, the isotope data from the BJD and the Qilian Mountain do not support the hypothesis regarding the Qilian Mountain being the recharge area for groundwater in the BJD. The evaporation line EL$_3$, which was obtained through linear regression of all available groundwater and lake water isotope data from the BJD, is very close to the local evaporation line EL$_1$ obtained from the evaporation experiments in this study. However, water samples from the Qilian Mountain are all plotted above the evaporation lines in the BJD (i.e., EL$_1$ and EL$_3$ in Fig. 5a). In particular, the meltwater and groundwater samples from the Qilian Mountain have $\delta$D and $\delta^{18}$O values higher than those determined by the intersection of the GMWL and local evaporation lines (Fig. 5a). Furthermore, the *d*-excess values of various waters in the Qilian Mountain are significantly higher than those of waters in the BJD and deviate significantly from the evaporation trend line for the study area (Fig. 5b). If groundwater in the BJD came from the Qilian Mountain as fast and deep subsurface flows in large fractures or faults underground as suggested by some studies (Chen et al., 2004), evaporation impacts would have been negligible, and the $\delta^{18}$O, $\delta$D and *d*-excess values of groundwater in the BJD would be similar to those of the source waters in the Qilian Mountain.

However, this is not the case (Fig. 5). Thus, the stable isotope data do not provide support for the remote-source hypothesis proposed by Chen et al., (2004) and Ding and Wang (2007).

*Insert Figure 5*

Figure 5a also shows that the shallow groundwater samples from the Yabulai Mountain area fall above the evaporation lines in the BJD, but follow a trend line (EL Yabulai: $y=4.2x-24.1$) that extends through some of the lakes (Fig. 5a). This suggests that shallow groundwater in the Yabulai Mountain maybe a source for some of the lakes in the BJD or the evaporation-infiltration conditions in the mountain areas were different from those in the desert. Two deep groundwater samples from Xugue and Gurinai (144m deep) on the northwestern and eastern margins of the BJD (Gates et al., 2008a) have similar $\delta^{18}O$ and $\delta D$ values to the groundwater in the BJD (Fig. 5a). Because of the limited number of deep groundwater samples, it is not yet possible to determine the relationship between them, but these aquifers may be connected as suggested by their similar isotope compositions (Fig. 5a).

### 5.3 Origin and residence time of groundwater

Based on the hydrogeological conditions and the water isotope data, we propose a new conceptual model for the origination of water in the BJD, which is schematically shown in Figure 6. The sources of groundwater and lake water in the BJD are mainly derived from meteoric precipitation in the desert and adjacent mountains. These mountains include the Zongnai Mountain in the east, the Yabulai Mountain in the southeast, and the Beida Moutain in the South, providing lateral groundwater flow toward the lakes in the BJD, especially the fresh and brackish lakes near the margin of the desert. Groundwater feeding most of the lakes, particularly the ones located farther away from the mountains in the BJD, however, receives at least partial recharge from infiltration water originated from local precipitation on the highly permeable sand dunes in the area. Similar results were reported from arid central Australia (Tweed et al., 2011). Based on the stable isotope ($\delta^{18}O$ and $\delta^2H$) data, Tweed et al. (2011) suggested that groundwater in that arid region was principally recharged during larger and intense rainfall events, but over longer timeframes, groundwater recharge was predominantly via diffuse processes rather than infiltration of floodwaters, even though the recharge may locally vary with distance from the floodplain. Our data show that the *d*-excess value of water decreases progressively from precipitation, infiltration, groundwater and lake water due to evaporation (Figs. 3b & 5b). For groundwater originated from the mountain areas, the *d*-excess value is generally higher than that originated from the local sand dunes because of more rapid infiltration and less evaporation under cooler and higher precipitation conditions in the mountains. The groundwater originated from local precipitation, on the other hand, inherits the *d*-excess value of the infiltration water in the dunes, which depends on the long-term environmental conditions in the area. Evolution of the lakes from brackish to saline also reflects the long-term accumulation of salts (Gong et al., 2016) and increase in the *d*-excess driven by lake surface evaporation.

*Insert Figure 6*

Only a few other studies also reported *d*-excess values and $\delta^{18}O$ values of groundwater from similar arid areas, such as Lake Eyre Basin (LEB), Australia (Tweed et al., 2011), and Jabal Hafit mountain in the United Arab Emirates (UAE) (Murad and

Mirghni, 2012). Analysis of these previously published *d*-excess values and $\delta^{18}O$ values of groundwater from these arid areas also reveals strong relationships between the two (Fig. 7), suggesting similar recharge processes as observed in the BJD. This implies that previous interpretation in terms of the origin of groundwater may need to be revised. For example, Wood (2010) interpreted the negative *d*-excess (−34.27‰ ~ −10.8‰) values of paleo-groundwater as indicative of influx of evaporated runoff into the Red Sea during the last wet period resulting in the negative *d*-excess values in the moisture source. However, the strong relationship between the *d*-excess and $\delta^{18}O$ values of Gachsaran aquifer indicates that the water was affected by evaporation.

*Insert Figure 7*

Assuming that water in the lakes was originated from the local precipitation in the desert, the water balance of the lakes can be analyzed using the following mass-balance equation:

$$A_{lake}(E_0 - P) = \alpha(A_{zone} - A_{lake})P \qquad (3)$$

where $A_{lake}$ and $A_{zone}$ are the total water surface area of the lakes and the catchment (or recharge) area for the lakes in the desert, respectively; $E_0$ is the mean annual potential evaporation of open water; $P$ is the mean annual precipitation; $\alpha$ denotes the ratio of total evaporative water loss from lakes to the total precipitation in the catchment. From Equation (3), the following relationship can be derived: required recharge area, $A_{zone}$, is controlled by the lakes and recharge ratio as:

$$\frac{A_{zone}}{A_{lake}} = 1 + \frac{1}{\alpha}\left(\frac{E_0}{P} - 1\right) \qquad (4)$$

The potential evaporation rate ($E_0$) of the lake water has been estimated as 1200-1550 mm/yr from the observation data of the Sumu Barun Jaran lake (Wang et al., 2014). Using $E_0$=1550 mm/yr (to account for the maximum effect of the evaporation loss) and P = 100 mm/yr, the estimated $E_0$/P ratio is 15.5. If the infiltration recharge of groundwater is only 5 mm/yr ($\alpha$=0.05), the recharge area that is required to balance the evaporative water loss from the lakes would be 291 time of the lakes area ($A_{zone}/A_{lake}$=291). Since the area of the lakes in the desert is about 22.3 km$^2$, the estimated catchment area $A_{zone}$ would be ~6489 km$^2$ in this scenario. As mentioned in Section 2, most of the lakes are concentrated in an area of ~2500 km$^2$, which is smaller than the estimated recharge area. The estimated recharge area with respect to $\alpha$=0.05, however, is much smaller than the area of the southeastern region of the BJD (~14,000 km$^2$). In a preliminary modeling simulation, Wang et al. (2014) suggested that the infiltration recharge would be 31 mm/yr. Using an $\alpha$ value of 0.31 the calculated $A_{zone}/A_{lake}$ ratio would be 47 and the required recharge area would decrease to ~1048 km$^2$, which is significantly less than the lake concentration area where most of lakes occur. Thus, it is highly possible that the evaporative water loss from the lakes could be balanced by local recharge in the desert area where lakes are located, without the need for external water sources.

The radiocarbon ages of DIC in groundwater are shown in Table 3, which range between 2 ka and 9 ka. This does not indicate very old water since groundwater ages in deserts are generally larger than 10 ka (Bentley et al., 1986; Kronfel et al., 1993; Sultan et al., 1997; Edmunds et al., 2006; Hagedorn, 2015). These ages do not represent the residence time of water due to the input of [14]C-deficient DIC derived from dissolution of old carbonates. Gates et al. (2008a) estimated that it would take 1-2 thousand years for groundwater to flow to the East Badain Lake from Yabulai Mountain (~25km apart) – the presumed

recharge area for groundwater in the BJD. The wells that we sampled in this study are located within 1 km of each other near the Sumu Jaran Lake (Fig. 1). However, the radiocarbon ages of DIC in groundwater varied greatly from 2560 yr BP at location WS-3 to 8250 yr BP at WS-1 (Fig. 8). Although both the tritium and $^{14}$C activities in groundwater decreased progressively along the flow path from the sand dune to the lake (Fig. 8), the rate of $^{14}$C change is much greater than that of tritium. The

$HCO_3^-$ in groundwater also increased along the flow path from WS-4 to WS-1 (Table 3; Figs. 1C & 8). Thus, the old $^{14}$C ages are most likely the result of the addition of old DIC or $^{14}$C-deficient DIC from the dissolution of old carbonates in the aquifer. The DIC of groundwater is likely composed of two sources: soil $CO_2$ in the recharge area and carbonates in the aquifer. The $\delta^{13}$C of DIC derived from dissolution of soil $CO_2$ in the recharge zone can be estimated using equilibrium isotope fractionation factors for carbonate-water system (Deines et al., 1974). The desert is sparsely vegetated with shrubs and grasses. Although

one C4 grass (*Agriophyllum squarrosum*) and two C4 shrubs (*Haloxylon ammodendron* and *Calligonum alaschanicum*) have been found the desert, the biomass in the region is dominated by C3 plants including *Caragana korshinskii* (C3), *Pugionium cornutum* and *Psammochloa villosa* (Yan et al., 2001; Wang et al., 2007; Ramawat, 2009). The shrubs are distributed on the dunes, but the lowland areas near the lakes are covered by grasses. The $\delta^{13}$C of soil $CO_2$ in soils hosting dense $C_3$ vegetation is about −23‰ (Cerling et al., 1991). In soils with <60% vegetation cover, $\delta^{13}$C of soil $CO_2$ is > −21‰,

due to mixing with atmospheric $CO_2$ resulting from low soil respiration rates (Cerling et al., 1991; Quade et al., 1989). Assuming that the $\delta^{13}$C of soil $CO_2$ is −20‰, the pH of the infiltration water in the soil zone in the recharge area is 5.3 and carbonates in aquifer have a $\delta^{13}$C value +2‰. We can calculate the fraction (*F*) of soil $CO_2$-derived DIC in groundwater from the measured DIC- $\delta^{13}$C value using the following mass balance relationship:

$$F = \frac{\delta^{13}C - \delta^{13}C_{carb}}{\delta^{13}C_{DIC-CO_2(soil)} - \delta^{13}C_{carb}} \qquad (5)$$

where the $\delta^{13}C_{carb}$ is $\delta^{13}$C of DIC derived from dissolution of carbonates in the aquifer, and $\delta^{13}C_{DIC-CO_2(soil)}$ represents $\delta^{13}$C of DIC derived from soil $CO_2$ in the recharge zone.

For example, the calculated *F* value for sample WS-1 is 58%. This suggests that ~58% of the DIC in this sample was derived from soil $CO_2$, and the radiocarbon age (8250 yr B.P.) of this sample could be a result of 42% dilution by DIC derived from dissolution of 20000 year old carbonates in the aquifer (Table 3). That demonstrates that just a small amount of DIC from the

dissolution of old carbonates can yield an erroneous DIC radiocarbon age that could be several thousand years (or more) too old. Various types of carbonate have been found in the lake area including tufa deposits, lacustrine carbonates and calcareous cementation (Yang et al., 2003). These carbonates provide possible sources of old DIC in groundwater.

*Insert Figure 8*

Furthermore, these groundwater samples contained detectable amounts of tritium, indicating at least some component of

modern recharge (Fig. 8; Table 3). This is because tritium is a radioactive isotope with a short half-life of 12.43 years and its concentration in a sample should fall below detection limit after 7-10 half-lives (~100 years). Tritium concentration in the atmosphere was elevated considerably during the nuclear era but has been declining steadily toward natural background levels after the test ban agreement in 1963 (Clark and Fritz, 1997). The yearly average tritium concentrations of precipitation

measured at the IAEA-GNIP station in Zhangye in the region have decreased from 101 TU in 1986 to 17 TU in 2003. Gates et al. (2008a) found significant amounts of bomb tritium in groundwater samples collected in 2004 and 2005 from Yabulai Mountain and the BJD, also indicating recent recharge by meteoric precipitation. However, it is important to note that the number of $^{14}$C and $^{3}$H samples that have been analyzed from the BJD is very limited and less than 10. Analyses of more groundwater samples and application of innovative dating techniques are required in order to better constrain the residence time of groundwater in the region. Our limited data show that radiocarbon dating of DIC in groundwater in the unconfined aquifer in the BJD is unreliable as dissolution of old carbonates in area contributes old DIC to groundwater, leading to erroneous old DIC radiocarbon ages (Table 3).

## 6 Summary and conclusions

Isotopic analyses of water samples from natural water bodies (including groundwater and lake water) as well as from evaporation experiments provide new insights into several key aspects of the water cycle in the Badain Jaran Desert:

1.  The lakes in the area all had significantly higher $\delta^2$H and $\delta^{18}$O but lower $d$-excess values than groundwater. The isotopic patterns confirm the previous findings that groundwater supplies the lakes in the sand dunes area and the lake water is well mixed in the vertical direction.

2.  The negative $d$-excess values of groundwater and lake water in the BJD are the result of intense evaporation. The evaporation experiments show that the water $d$-excess value decreases with evaporation in the desert. Thus, the $d$-excess value is expected to decrease continuously from the precipitation to groundwater and lakes if the groundwater and lake water were evolved from the same meteoric precipitation.

3.  The groundwater in the unconfined aquifer in the desert was derived primarily from modern meteoric precipitation in the region and possible recharge areas include the eastern and southeastern areas of the BJD and the adjacent mountains to the east and south of the desert.

4.  The radiocarbon ages of DIC in groundwater in the BJD do not represent the residence time of water in the unconfined aquifer and are too old due to the addition of old DIC from the dissolution of ancient carbonates in the aquifer. Although the residence time of groundwater remains to be determined, the variation patterns in $\delta^{18}$O, $\delta^2$H and $d$-excess values of lake water, groundwater, and precipitation in the region, along with the presence of detectable amounts of tritium, suggest that the average residence time is on the order of decades. Understanding the origin, flow path and residence time of groundwater in the region is essential for sustainable management of the water resources.

This study provides direct evidence showing that evaporation is an important factor influencing the isotope composition of infiltration water and needs to be considered when using the deuterium and oxygen isotopes to infer water resources of semi-arid and arid regions. This study also demonstrated that the characteristic water isotopic patterns resulting from evaporation could be utilized to help resolve ambiguities in the interpretation of water isotope data in terms of recharge sources, especially, in the arid regions, such as the central Australia and the deserts of United Arab Emirates.

**Author contribution**: Xu-sheng Wang designed and carried out the field experiments. Bill X. Hu studied the water circle system in the study area. Xiujie Wu and Yang Wang analyzed the data and interpreted the results. Xiujie Wu prepared the manuscript with contributions from all co-authors.

5 **Competing interests**: Author Bill X. Hu is a member of the editorial board of the journal.

**Acknowledgment**: This research was sponsored by National Science Foundation of China (No. 91125024). Xiujie Wu received financial support from the China Scholarship Council (CSC). Stable isotope analysis was performed at the National High Magnetic Field Laboratory, which is supported by U.S. National Science Foundation Cooperative Agreement No. DMR-1157490 and the State of Florida.

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

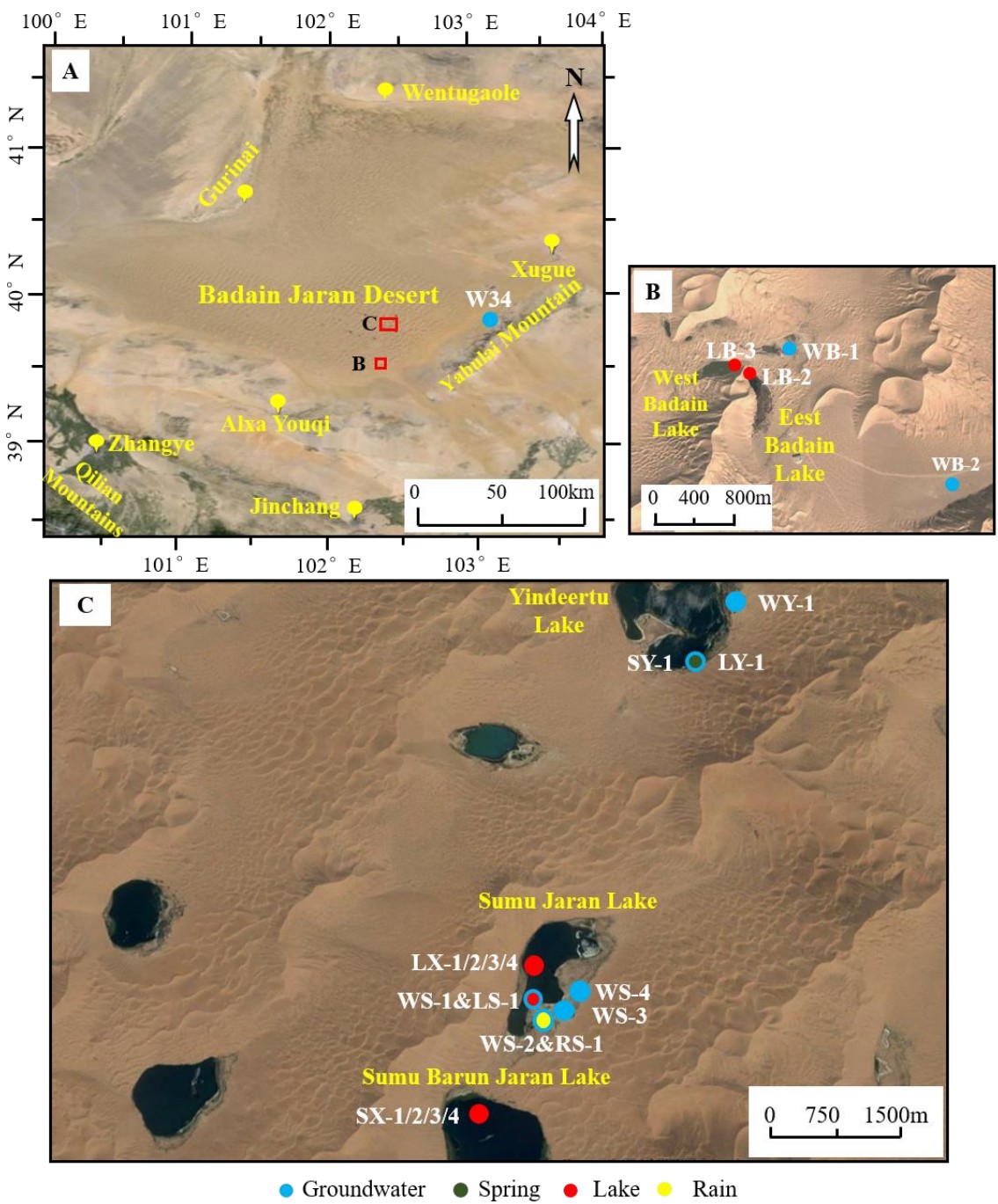

**Figure 1: Maps showing location of the Badain Jaran Desert (A), the Badain lake sampling area (B) and the Sumu Jaran lake sampling area (C). W34 in (A) is sampling site of Gates et al. (2008a).**

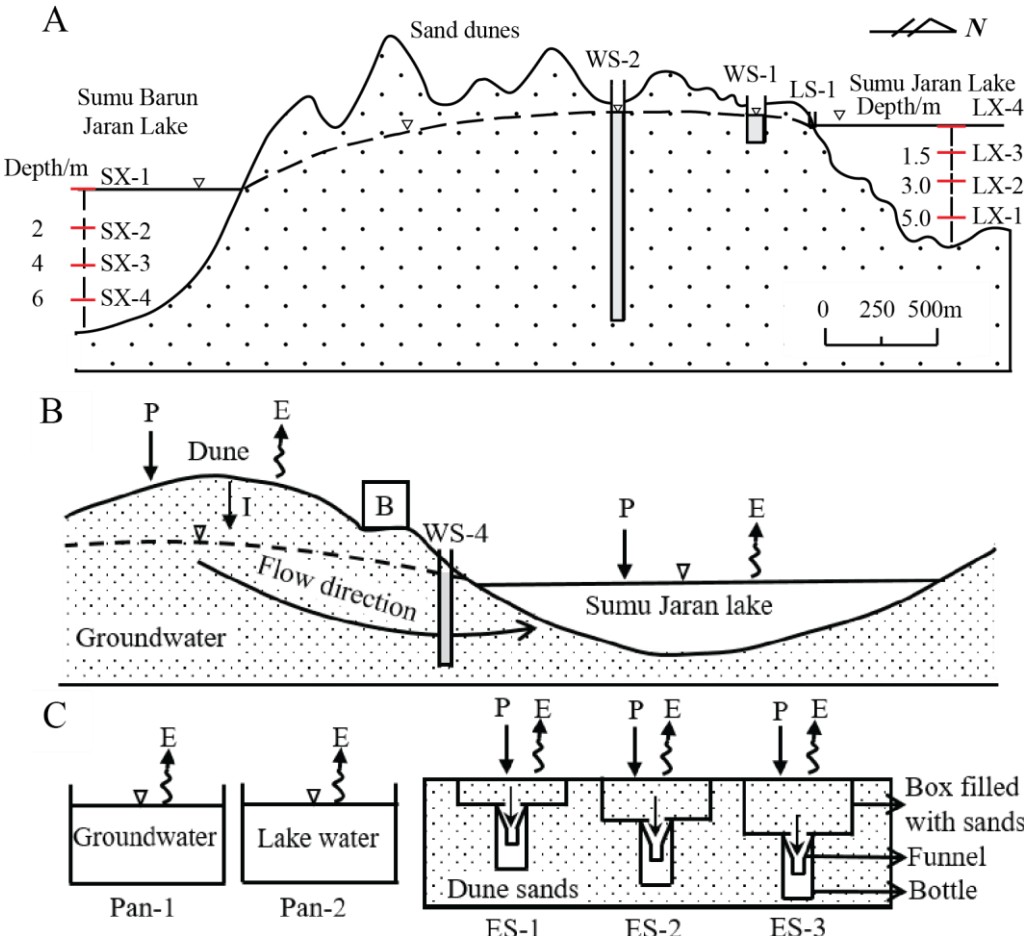

**Figure 2: Schematic diagram showing the cross-section profile between the Sumu Jaran Lake and the Sumu Baran Jaran Lake as well as the water sampling points (A) and the groundwater flow direction (B) and the evaporation experiments (C) in the Sumu Jaran Lake area. P denotes the precipitation, E is the evaporation and I is the infiltration. Pan-1 and Pan-2 are initially filled with groundwater from WS-4 and lake water from the Sumu Jaran Lake, respectively.**

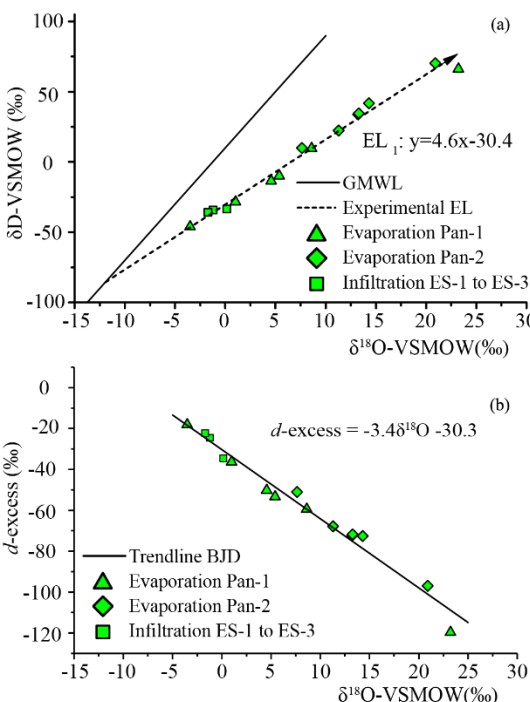

**Figure 3: The relationship between δD and δ$^{18}$O (a) and between $d$-excess and δ$^{18}$O (b) of water samples from evaporation experiments.**

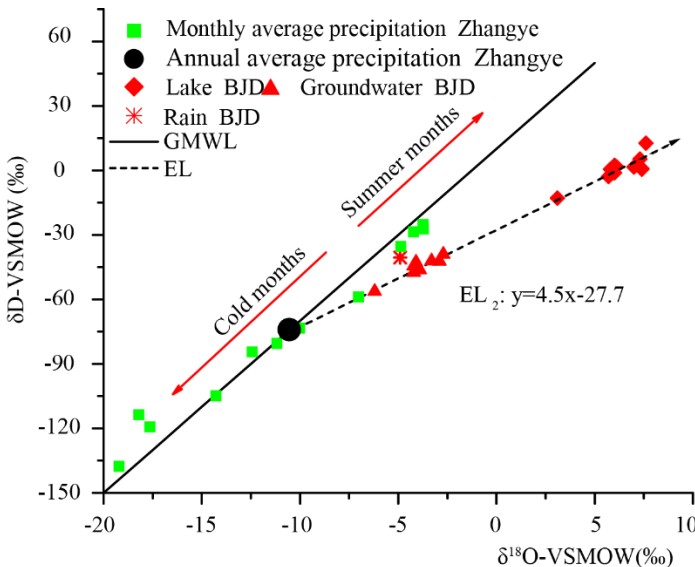

**Figure 4: The δD vs. δ¹⁸O plot of natural groundwater, lake water, and precipitation in the desert. Also shown are weighted monthly average and weighted annually average isotope ratios of precipitation at the IAEA-GNIP station in Zhangye.**

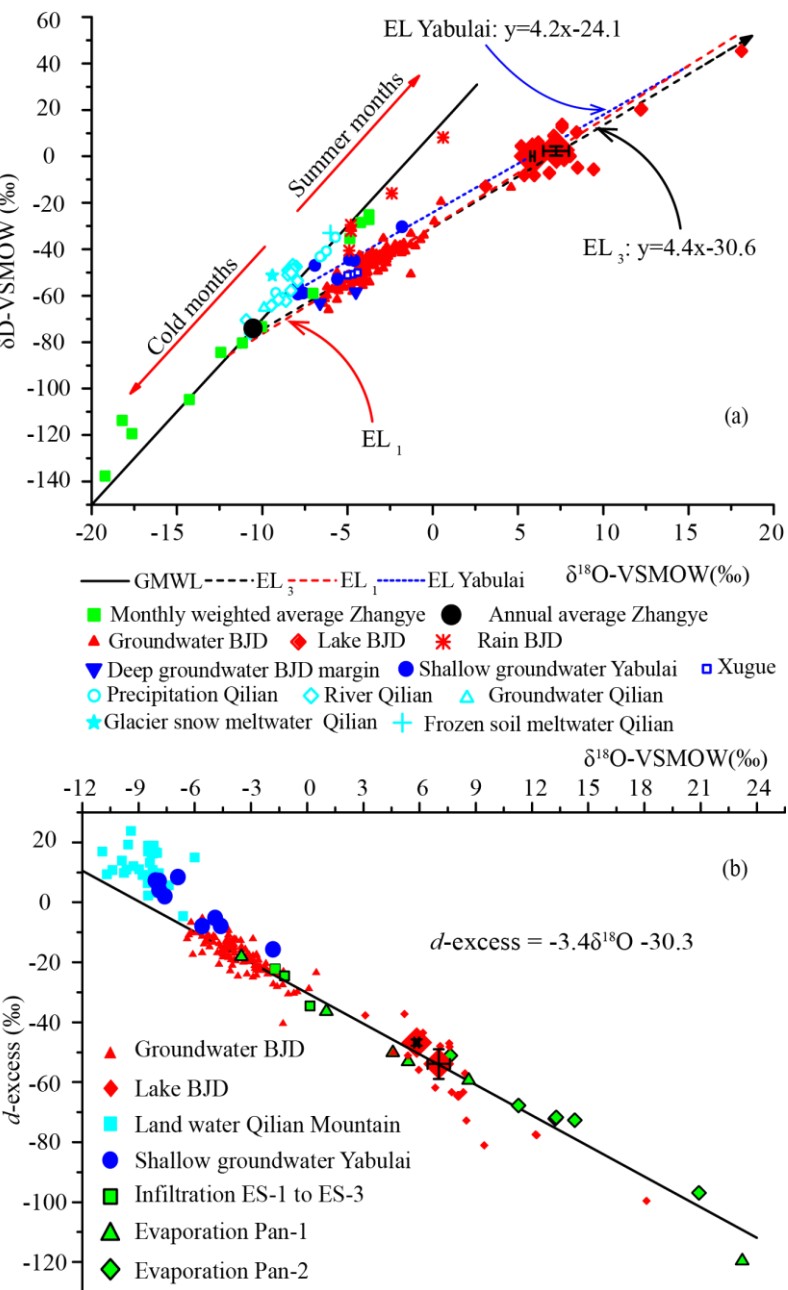

**Figure 5: The plot of δD vs δ¹⁸O values (a) and *d*-excess vs δ¹⁸O values (b) of groundwater and lake water samples from the BJD (red symbols), including new data from his study and previously published data from the literature (Gates et al., 2008a; Zhang et al., 2011; Zhao et al., 2012). The trend line in (b) is established from our evaporation experiments (Fig.4b). Also shown are the isotope data from the Qilian Mountain area (light blue symbols) for comparison. The two larger diamond dots with black cross inside are the average values with error bar for the Sumu Jaran and Sumu Badain Jaran lakes sampled at different depths. Isotope data for deep groundwater in Gurinai and Xugue and shallow groundwater in Xugue and Yabulai Mountains in (A) are from Gates et al. (2008a). The water isotope data for the Qilian Mountains include precipitation (Wu et al., 2010; Chen et al., 2012), and land water including groundwater (Li et al., 2016), rivers (average for each river) (Chen et al., 2012; Li et al., 2016), glacier snow melt water and frozen soil melt water (Li et al., 2016).**

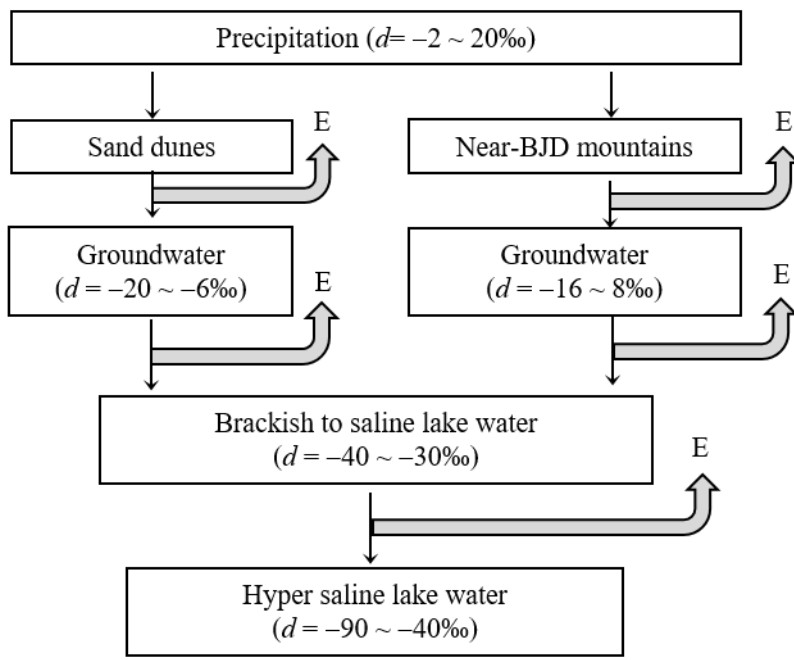

**Figure 6: The conceptual model of the d-excess changing routines in the BJD. E represents evaporation.**

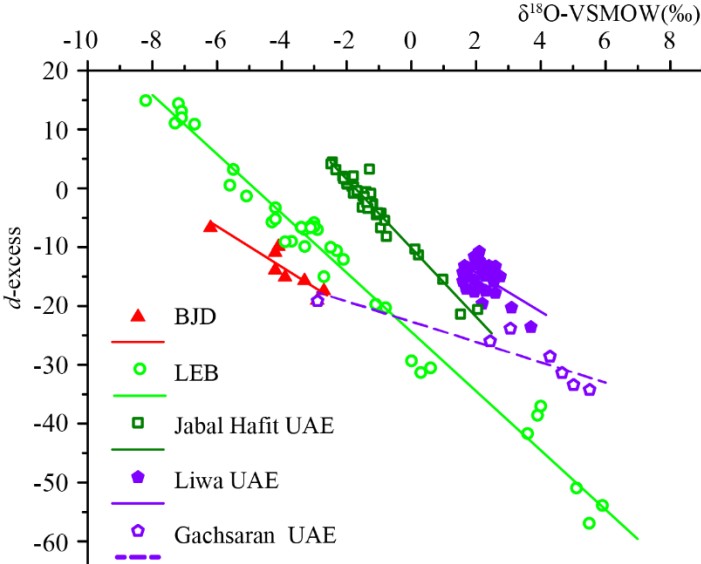

**Figure 7: Comparison of *d*-excess and δ¹⁸O values of groundwater samples from the BJD (red triangle), Lake Eyre Basin, Australia (green circle) (Tweed et al., 2011), Jabal Hafit mountain, UAE (dark green square) (Murad and Mirghni, 2012), and the two aquifers, Liwa and Gachsaran of Rub Al Khali, UAE (purple) (Wood, 2010). The trend lines are established and plotted in same color following the data.**

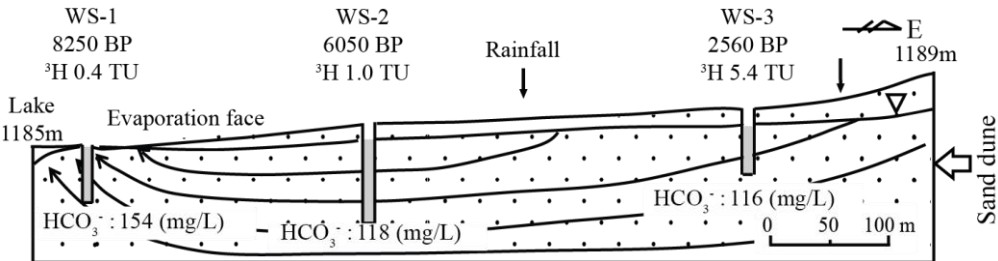

**Figure 8: The flow model of groundwater near the Sumu Jaran Lake. The HCO₃⁻ concentrations of each wells are shown in the figure.**

Table. 1 Results from the pan-evaporation experiments

| Pans | Number of days | Residual water volume percentage | $\delta^{18}O_{V\text{-}SMOW}$ (‰) | $\delta D_{V\text{-}SMOW}$ (‰) | $d$-excess (‰) |
|------|----------------|----------------------------------|-----------------------------------|-------------------------------|----------------|
| Pan-1 | 0 | 100 | –3.5 | –46 | –18 |
| | 1 | 83 | 1.0 | –28 | –36 |
| | 2 | 52 | 4.6 | –14 | –50 |
| | 3 | 41 | 5.4 | –10 | –53 |
| | 4 | 37 | 8.6 | 10 | –59 |
| | 5 | 19 | 23.2 | 66 | –120 |
| Pan-2 | 0 | 100 | 7.6 | 10 | –51 |
| | 1 | 83 | 11.3 | 22 | –68 |
| | 2 | 66 | 13.2 | 34 | –72 |
| | 3 | 64 | 13.3 | 35 | –71 |
| | 4 | 54 | 14.3 | 42 | –72 |
| | 5 | 30 | 18.2 | 70 | –76 |

Table. 2 The sampling location and data and EC, $\delta^{18}O$ and $\delta D$ values of water samples from lakes, wells and spring in the BJD.

| Sample identity | Category | Sampling date | Altitude(N) | Longitude (E) | EC (mS/cm) | $\delta^{18}O_{V\text{-}SMOW}$ (‰) | $\delta D_{V\text{-}SMOW}$ (‰) | $d$-excess (‰) |
|---|---|---|---|---|---|---|---|---|
| RS-1 | rain | 2012-08-08 | 39°48.213' | 102°25.567' | | −4.9 | −41 | −2 |
| WS-1 | well | 2012-08-06 | 39°48.335' | 102°25.490' | 0.6 | −6.2 | −56 | −6 |
| WS-2 | well | 2012-08-06 | 39° 8.213' | 102°25.567' | 0.5 | −4.2 | −48 | −14 |
| WS-3 | well | 2012-08-06 | 39°48.245' | 102°25.768' | 0.3 | −2.9 | −42 | −19 |
| WS-4 | well | 2012-08-06 | 39°48.374' | 102°25.869' | 0.6 | −2.7 | −39 | −17 |
| WY-1 | well | 2012-09 | 39°51.273' | 102°27.182' | 0.6 | −3.3 | −42 | −16 |
| WB-1 | well | 2012-08-08 | 39°33.278' | 102°22.205' | 0.9 | −4.2 | −45 | −11 |
| WB-2 | well | 2012-09 | 39°32.368' | 102°23.388' | | −4.1 | −43 | −10 |
| SY-1 | spring | 2012-08-06 | 39°50.800' | 102°26.808' | 0.7 | −3.9 | −46 | −15 |
| LY-1 | lake | 2012-08-06 | 39°50.800' | 102°26.808' | 230.6 | 6.1 | 2 | −47 |
| LS-1 | lake | 2012-08-06 | 39°48.338' | 102°25.473' | 178.6 | 6.0 | 2 | −46 |
| LX-4 | lake | 2012-08-07 | 39°48.569' | 102°25.489' | 227.0 | 7.0 | 2 | −54 |
| LX-3 | lake | 2012-08-07 | 39°48.569' | 102°25.489' | 228.0 | 7.3 | 5 | −53 |
| LX-2 | lake | 2012-08-07 | 39°48.569' | 102°25.489' | 324.7 | 7.4 | 1 | −59 |
| LX-1 | lake | 2012-08-07 | 39°48.569' | 102°25.489' | 548.9 | 7.4 | 1 | −59 |
| SX-1 | lake | 2012-08-07 | 39°47.561' | 102°25.095' | 223.0 | 6.0 | −1 | −49 |
| SX-2 | lake | 2012-08-07 | 39°47.561' | 102°25.095' | 227.2 | 5.9 | −1 | −48 |
| SX-3 | lake | 2012-08-07 | 39°47.561' | 102°25.095' | 228.2 | 5.7 | −3 | −49 |
| SX-4 | lake | 2012-08-07 | 39°47.561' | 102°25.095' | 179.2 | 5.8 | 1 | −45 |
| LB-2 | lake | 2012-08-08 | 39°33.110' | 102°21.845' | 3.6[#] | 3.1 | −13 | −38 |
| LB-3 | lake | 2012-08-08 | 39°33.182' | 102°21.748' | 796.4[#] | 7.6 | 13 | −48 |

"#" data are cited from Yang and Williams, 2003, they were given in TDS (g/l) and roughly transformed to EC (mS) through multiplying by 2.

Table. 3 Radiocarbon and $^3$H contents of groundwater samples

| Sample identity | $^3$H (TU) | Water T (°C) | pH | Uncorrected age (yr BP) | $^{14}$C pMC | Measured DIC-$\delta^{13}$C (‰) | q | Estimated age of carbonate-derived DIC |
|---|---|---|---|---|---|---|---|---|
| WS-1 | 0.4±0.4 | 18.4 | 8.3 | 8250±40BP | 35.81 | −11.1 | 0.58 | 19614 |
| WS-2 | 1.0±0.5 | 16.0 | 8.4 | 6050±40BP | 47.09 | −6.7 | 0.38 | 9835 |
| WS-3 | 5.4±0.7 | 16.0 | 8.3 | 2560±40BP | 72.71 | −8.2 | 0.45 | 4665 |
| WS-4 | 5.2±0.7 | 16.9 | 8.3 | 3990±40BP | 60.85 | −7.3 | 0.41 | 6778 |
| WB-1 | | 18.4 | 8.3 | 2840±30BP | 70.22 | −7.5 | 0.42 | 4898 |
| WB-2 | | 18.4 | 8.3 | 2940±30BP | 69.35 | −8.3 | 0.46 | 5400 |
| WY-1 | 2.8±0.6 | 17.5 | 8.4 | 4210±40BP | 59.21 | −8.8 | 0.48 | 8060 |