# Peer review of "Origin of water in the Badain Jaran Desert, China: New insight from isotopes"

_Hydrology and Earth System Sciences, 2016_

## Referee Comment (RC1) · Anonymous Referee #1 · 6 Apr 2017

Comments to the Authors

The manuscript on "Water resources in the Badain Jaran Desert, China: New insight from isotopes" by Xiujie Wu, et al. is providing evidence that groundwater in the Badain desert is locally recharged and consists of young water based on stable isotope samples collected from groundwater wells, lakes and during evaporation experiments. The authors argue against opinions published earlier (e.g., Chen et al. 2004, Gates et al. 2008). Beside stable isotopes, a main argument against old groundwater originating from the Qilian Mountains – as proposed earlier - is that DIC based carbon-14 ages are impacted by old carbonates. The authors argue that evaporative enrichment of stable isotopes and the extrapolation of the evaporation line back to source water on the LMWL proves that recharge occurs locally. The manuscript is covering a very interesting topic and research site, it is well written and structured. The collected data (two

campaigns, 21 stable isotope samples, 7 carbon-14 groundwater ages) is combined with data from earlier studies of other authors. The material collected for this work seems satisfying but the authors could improve the manuscript, especially the methods chapter and their presentation of figures and tables. I recommend to accept the paper for HESS after major revisions. My comments given below aim for an additional improvement of the manuscript.

General comments

- Since water resources are the main topic/title and there were plans to use these resources for a large water diversion project (Chen et al., 2004) in a very sensitive (arid) environment, a better description of hydrological components and an overall balance would be helpful. Would it be possible to calculate the recharge area that has to feed an evaporation loss of the lakes given? It would be interesting for the readers to get a better description of the hydrogeology and aquifer characteristics in the area (unconfined aquifer, page 11, line 3).

- Are the evaporation experiments and especially the pan size that were used representative for real evaporation processes? How were the pans constructed and installed? - Metal rings – e.g., in comparison to "class-A-evaporation pan" recommendations.

- The given evaporation lines should be directly compared and values discussed with those of other studies (e.g., Wu et al. 2014, Chen et al., 2004). Because a main argument for source water relies on an extrapolated value of the LMWL it would be necessary to provide best evidence for this value.

- The method section lacks precise description and detailed information (e.g., on conducted 14C corrections, gas preparation methods for stable isotopes). Would it be possible to correct for the described carbonate contribution based on measured values? Would it be possible to use DOC for 14C-dating or other dating approaches? Was hydrochemistry data evaluated from the collected samples as well?

Please find more detailed comments attached.

Please also note the supplement to this comment:
http://www.hydrol-earth-syst-sci-discuss.net/hess-2016-692/hess-2016-692-RC1-supplement.pdf

**Supplement:**

**Comments to the Authors**

The manuscript on "*Water resources in the Badain Jaran Desert, China: New insight from isotopes*" by Xiujie Wu, et al. is providing evidence that groundwater in the Badain desert is locally recharged and consists of young water based on stable isotope samples collected from groundwater wells, lakes and during evaporation experiments. The authors argue against opinions published earlier (e.g., Chen et al. 2004, Gates et al. 2008). Beside stable isotopes, a main argument against old groundwater originating from the Qilian Mountains – as proposed earlier - is that DIC based carbon-14 ages are impacted by old carbonates. The authors argue that evaporative enrichment of stable isotopes and the extrapolation of the evaporation line back to source water on the LMWL proves that recharge occurs locally. The manuscript is covering a very interesting topic and research site, it is well written and structured. The collected data (two campaigns, 21 stable isotope samples, 7 carbon-14 groundwater ages) is combined with data from earlier studies of other authors. The material collected for this work seems to be satisfying but the authors could improve the manuscript, especially the methods chapter and their presentation of figures and tables. I recommend to accept the paper for HESS after major revisions. My comments given below aim for an additional improvement of the manuscript.

**General comments**

- Since water resources are the main topic/title and there were plans to use these resources for a large water diversion project (Chen et al., 2004) in a very sensitive (arid) environment, a better description of hydrological components and an overall balance would be helpful. Would it be possible to calculate the recharge area that has to feed an evaporation loss of the lakes given? It would be interesting for the readers to get a better description of the hydrogeology and aquifer characteristics in the area (unconfined aquifer, page 11, line 3).
- Are the evaporation experiments and especially the pan size that were used representative for real evaporation processes? How were the pans constructed and installed? - Metal rings – e.g., in comparison to "class-A-evaporation pan" recommendations.
- The given evaporation lines should be directly compared and values discussed with those of other studies (e.g., Wu et al. 2014, Chen et al., 2004). Because a main argument for source water relies on an extrapolated value of the LMWL it would be necessary to provide best evidence for this value.
- The method section lacks precise description and detailed information (e.g., on conducted $^{14}$C corrections, gas preparation methods for stable isotopes). Would it be possible to correct for the described carbonate contribution based on measured values? Would it be possible to use DOC for 14C-dating or other dating approaches? Was hydrochemistry data evaluated from the collected samples as well?

**Specific comments**

Title:
- Use "Groundwater studies …" instead of "Water resources …", otherwise your work should focus more on hydrological budget quantification and hydrogeological aspects.

Introduction:
- Page 2, lines 31-33: This sentence is summarizing results and would fit better into the conclusion or abstract section.

Methods:
- You mention the GNIP station Zhangye. Please add the distance in km to the study site and further information on time sampled, number of samples used for LMWL.
- IAEA/WMO, the internet link should be given as a reference in the references section. See also recommendations for referencing to GNIP data on the WISER database at IAEA.
- Page 5, line 6: "…artificial rainfall with 250 mL in 6 min …" It would be more informative to provide irrigation intensities in mm/min for 6 min.
- Page 5, line 13: I would recommend "Isotope analyses" or "Laboratory methods" instead of chemical analyses, because hydrochemistry is not discussed and isotope methods are no chemical methods.
- Page 5, line 17: "…Five groundwater samples …" In Table 3 seven ages are given for groundwater!?
- Page 5, line 24. For the stable isotope analysis please give the specific gas preparation methods that were used, e.g., Gasbench, H-device, or TCEA?
- Page 5, line 26. Please use appropriate definition of delta values. $R_{SA}/R_{ST}$ and not $R_{V\text{-}SMOW}$. This is especially important because you also give $d^{13}C$ values in Table 3. These are not defined against V-SMOW but VPDB I guess!
- Your precision is given as 1‰ and 0.1‰ for $d^2H$ and $d^{18}O$ respectively. Your d-excess results therefore should not be given with commas (see values in Tables as well). What is the precision for your $d^{13}C$ values? What are the precision of your $^3H$ values? Did you test any post corrections for $^{14}C$ DIC?

Results and discussions
- Page 6, line 10-14: Please give d-excess values without digits.
- Page 6, line 11: Why did you distinguish between groundwater and lake water? Please discuss results!
- Page 6, line 22: You show that d-excess values are negatively correlated with $d^{18}O$ values (Figures 4b, 6b). Please discuss what this exactly means in your case. Usually these plots are used to argue for water vapor origin.
- Page 6, line 27: "… as Fraction Modern (Fmdn)…" Usually given in percent modern carbon as pMC. See also Table 3.
- Page 8, line 10: "… from nearby IAEA GNIP ..." Please provide information on distance and elevation of the station.
- You do not describe and discuss field parameters EC given in Table 2.
- The discussion on 14C free carbonate contribution to DIC is vague. Figure 9 is difficult to understand.

References:
Please point out all Chinese references (in Chinese) for the international readers that do not understand Chinese language.

*Figures and Tables:*
- Figure 1. Please include location of GNIP station Zhanye in Fig. 1A)
- Figures 2 and 3 could be merged together
- Figure 6: The two diamond dots are not clearly visible, not visible in 6b. "Land water" should be rephrased! Soil water?
- Figure 7: E – fluxes are misleading! Evaporation from groundwater to lake water fluxes could be better placed at boxes.
- Table 1: d-excess values without digit.

- Table 2: Category should be rephrased into type. EC is given in mS not Ms! d18O and d2H measured against VSMOW. d-excess values without digit.
- Table 3: Temperature without digits or consistently. d13C against VPDB.

**Technical corrections**

- Page 1, line 11: "(d$^2$H-d$^{18}$O) instead of (=d$^2$H...)
- Page 1, line 25: "… are scarce in arid regions, due …" Please delete "in arid regions"
- Page 2, line 5: progress instead of progresses.
- Page 3, line 3:"… to104° …" Space is missing.
- Page 5, line 20. Beta instead of Bata.
- Page 5, line 20. Beta Analytic Inc. (Miami, Florida, USA).
- Page 6, line 22: " … and that it is negatively …" instead of "…and strongly and negatively .."

*Figures and Tables:*
- Table 2. Table caption should include all columns. Location, date and EC is not mentioned.
- Figure caption Figure 5: delete space between d18 and O.
- Figure caption Figure 6: Please correct figure caption (e.g., d18O).

---

## Referee Comment (RC2) · Anonymous Referee #2 · 12 Apr 2017

General opinion

This manuscript (hess-2016-692; Water resources in the Badain Jaran Desert, China: New insight from isotopesXiujie Wu, Xu-Sheng Wang, Yang Wang, and Bill X. Hu) is a concise presentation of the application of well-established isotope techniques in identification of groundwater sources for a group of desert lakes in China. A relatively small amount of isotopic data points to recent precipitation being the sole source of water in the coupled groundwater-surface water system. The isotopic evidence for that seems to be unequivocal, however, some questions (see specific comments) should be addressed in order to strengthen the line of reasoning. On the other hand, data on stable isotopic composition of DIC are not used at all and the part explaining 14C patterns requires some elaboration. Interpretation of presented data could be supported with a more detailed description of the hydrological

setting and limnology of the lakes. Text reads well, it is comprehensible and clearly presents the context, goals and conclusions of the study, however the description of the sampling campaign needs to be improved. My recommendation is to accept the manuscript after major revision. The specific and technical comments are attached.

Please also note the supplement to this comment:
http://www.hydrol-earth-syst-sci-discuss.net/hess-2016-692/hess-2016-692-RC2-supplement.pdf

**Supplement:**

**General opinion**

This manuscript (hess-2016-692; Water resources in the Badain Jaran Desert, China: New insight from isotopesXiujie Wu, Xu-Sheng Wang, Yang Wang, and Bill X. Hu) is a concise presentation of the application of well-established isotope techniques in identification of groundwater sources for a group of desert lakes in China. A relatively small amount of isotopic data points to recent precipitation being the sole source of water in the coupled groundwater-surface water system. The isotopic evidence for that seems to be unequivocal, however, some questions (see specific comments) should be addressed in order to strengthen the line of reasoning. On the other hand, data on stable isotopic composition of DIC are not used at all and the part explaining [14]C patterns requires some elaboration. Interpretation of presented data could be supported with a more detailed description of the hydrological setting and limnology of the lakes. Text reads well, it is comprehensible and clearly presents the context, goals and conclusions of the study, however the description of the sampling campaign needs to be improved.

My recommendation is to accept the manuscript after major revision.

**Specific comments**

1. Given that only few lakes were sampled a question arises, not addressed in the manuscript, if results of this study apply to all lakes of the Badain Jaran Desert. How big is the part of the desert where the lakes occur? It is not delineated on Fig. 1. Is the hydrogeological setting uniform throughout that part of the desert, so that the recharge – discharge pattern proposed in the manuscript might apply for the whole area?

2. Did conditions of evaporation during the evaporation experiments correspond to the period of infiltration (also for recharge areas in the mountains)?

3. (p. 2, line 25-27) Does this sentence refer to Qilian Mountains mentioned further in the discussion?

4. (p. 3, lines 8-9) What is the relevance of the information on the depth of root penetration?

5. Chapter 3.1 Field Sampling. This part lacks some detail on the study area and sampling procedures. How many lakes were sampled? Five, or also some other of the "various" lakes? Provide their surface areas, maximum depths and volumes if known and explain if these characteristics are typical for the region. Provide information on wells depths and screening intervals as this is important in evaluation of well samples. Are the wells regularly used? If not, then 20 minute long pumping (with a peristaltic pump?) might not be enough to obtain a representative water sample from the aquifer.

6. (p6, lines 23-25) Was the good vertical mixing of lake waters confirmed by measurements of temperature and conductivity?

7. (p. 9, lines 13-14) Are there any data on groundwater flow directions/hydraulic heads in the lake area to support this notion? Generally, the hydrogeological conditions are poorly explained in the manuscript.

8. (p.10, lines 5-14) The bicarbonate and $\delta^{13}C_{DIC}$ data do not support the gradual dilution of $^{14}C$ with the old inorganic carbon derived from carbonate dissolution along the entire groundwater pathway. Both parameters change significantly only between points WS2 to WS1 pointing to isotopic exchange as a cause of the apparent increase of radiocarbon ages between WS4 to WS2. This discussion would benefit from any quantitative considerations on the isotopic evolution of DIC along the transect. However, the conditions of carbonate dissolution and isotopic exchange are controlled by aquifer properties which are not described in the manuscript.

9. (p.10, lines 27-29) The vertical homogeneity of lake water implies only good vertical mixing. The (isotopic?) steady state is a separate issue and can be assessed only from the temporal patterns. Is there any evidence for lake waters being in the steady state? By the way, what causes vertical mixing of the lakes? Wind or diurnal water temperature fluctuations?

10. Figs. 1B and 8 are not consistent with respect to the relative positions of wells and the Sumu Jaran Lake. The lake is located between W3 and W2 in Fig. 1B and to the west of W1 in Fig.8.

**Technical comments**

(p. 1, line 26) please remove "arid regions"

(p. 7, line 22) "and ARE strongly"

Fig. 7. Should be "brackish" not "blackish"

---

## Author Comment (AC1) · 27 May 2017

Reviewer 1ïïjŽ General comments Since water resources are the main topic/title and there were plans to use these resources for a large water diversion project (Chen et al., 2004) in a very sensitive (arid) environment, a better description of hydrological components and an overall balance would be helpful. Would it be possible to calculate the recharge area that has to feed an evaporation loss of the lakes given? It would be interesting for the readers to get a better description of the hydrogeology and aquifer characteristics in the area (unconfined aquifer, page 11, line 3). Reply: We agree. According to the comment, two points are highlighted in the revised manuscript: (1) An analysis of the mean annual water balance is performed in Section 5.3. (see Page 10; Line 26-32 & Page 11; Line 1-13) (2) The characteristics of the aquifer system in the BJD are described in the second paragraph of Section 2. (see Page 3, Line 11-23) Are

the evaporation experiments and especially the pan size that were used representative for real evaporation processes? How were the pans constructed and installed? - Metal rings – e.g., in comparison to "class-A-evaporation pan" recommendations. Reply: The evaporation experiments were performed using plastic pans with relatively small size. We did not use the class-A-evaporation pan which is specially applied to measure the potential evaporation rate in weather stations, because the objective of this experiment is limited to check the varying isotope components of open water during evaporating. We have revised the text to clarify this issue (see Page 5, Line 6,7) The given evaporation lines should be directly compared and values discussed with those of other studies (e.g., Wu et al. 2014, Chen et al., 2004). Because a main argument for source water relies on an extrapolated value of the LMWL it would be necessary to provide best evidence for this value. Reply: Accept. We compare the evaporation lines in our study with previous studies in the discussion part of Section 5.2. (see Page 8, Line 28-33& Page 9, Line 1-3) The method section lacks precise description and detailed information (e.g., on conducted 14C corrections, gas preparation methods for stable isotopes). Would it be possible to correct for the described carbonate contribution based on measured values? Would it be possible to use DOC for 14C-dating or other dating approaches? Was hydrochemistry data evaluated from the collected samples as well? Reply: Additional information is presented in Section 3.3. (see Page 5, Line 32& Page 6, 1-13) We make a correction of the carbonate contribution for the groundwater age. (see Page 11, Line 26-33; Page 12, Line 1-13) The DIC of groundwater is likely composed of two sources: soil $CO_2$ in the recharge area and carbonates in the aquifer. The $\delta$13C of DIC derived from dissolution of soil $CO_2$ in the recharge zone can be estimated using equilibrium isotope fractionation factors for carbonate-water system (Deines et al., 1974). The desert is sparsely vegetated with shrubs and grasses. Although one C4 grass (Agriophyllum squarrosum) and two C4 shrubs (Haloxylon ammodendron and Calligonum alaschanicum) have been found the desert, the biomass in the region is dominated by C3 plants including Caragana korshinskii (C3), Pugionium cornutum and Psammochloa villosa (Yan et al., 2001; Wang et al., 2007; Ramawat,

2009). The shrubs are distributed on the dunes, but the lowland areas near the lakes are covered by grasses. The $\delta$13C of soil CO2 in soils hosting dense C3 vegetation is about $-23$‰ (Cerling et al., 1991). In soils with <60% vegetation cover, $\delta$13C of soil CO2 is $> -21$‰ due to mixing with atmospheric CO2 resulting from low soil respiration rates (Cerling et al., 1991; Quade et al., 1989). Assuming that the $\delta$13C of soil CO2 is $-20$‰ the pH of the infiltration water in the soil zone in the recharge area is 5.3 and carbonates in aquifer have a $\delta$13C value +2‰ We can calculate the fraction (F) of soil CO2-derived DIC in groundwater from the measured DIC- $\delta$13C value using the following mass balance relationship: F=($\delta$^13 C-$\delta$^13 C_carb)/($\delta$^13 C_(DIC-ãĂŰCOãĂŮ_2 (soil))-ãĂŰ $\delta$ãĂŮ^13 C_carb ) (5) where the $\delta$13Ccarb is $\delta$13C of DIC derived from dissolution of carbonates in the aquifer, and $\delta$^13 C_(DIC-ãĂŰCOãĂŮ_2 (soil)) represents $\delta$13C of DIC derived from soil CO2 in the recharge zone. For example, the calculated F value for sample WS-1 is 58%. This suggests that ∼58% of the DIC in this sample was derived from soil CO2 , and the radiocarbon age (8250 yr B.P.) of this sample could be a result of 42% dilution by DIC derived from dissolution of 20000 year old carbonates in the aquifer (Table 3). That demonstrates that just a small amount of DIC from the dissolution of old carbonates can yield an erroneous DIC radiocarbon age that could be several thousand years (or more) too old. Various types of carbonate have been found in the lake area including tufa deposits, lacustrine carbonates and calcareous cementation (Yang et al., 2003). These carbonates provide possible sources of old DIC in groundwater. Unfortunately we were unable to do the 14C-dating using DOC or any other dating approaches, do not have other hydrochemistry data for the collected samples.

Specific comments

Title: Use "Groundwater studies . . ." instead of "Water resources . . .", otherwise your work should focus more on hydrological budget quantification and hydrogeological aspects. Reply: We revise the title to "Origin of water . . ." because this study is focused to reveal the origination of water in the BJD. (see Page 1, Line 1) Introduction: Page 2,

lines 31-33: This sentence is summarizing results and would fit better into the conclusion or abstract section. Reply: Accept. I find this sentence in the conclusion part, so I delete this sentence here. Methods: You mention the GNIP station Zhangye. Please add the distance in km to the study site and further information on time sampled, number of samples used for LMWL. Reply: Accept. Zhangye is the nearest IAEA-GNIP (International Atomic Energy Agency Global Network of Isotopes in Precipitation) station, approximately 170 km to the site-B in the BJD (Fig. 1A) at the altitude of ∼1400 m, with available data of IAEA/WMO for the period from 1986 to 2003. (see Page 4, Line 28, 29) IAEA/WMO, the internet link should be given as a reference in the references section. See also recommendations for referencing to GNIP data on the WISER database at IAEA. Reply: Accept. (see Page 15, Line 11) Page 5, line 6: ". . .artificial rainfall with 250 mL in 6 min . . ." It would be more informative to provide irrigation intensities in mm/min for 6 min. Reply: Accept. The mean "rainfall" intensity is 0.167 mm/min (see Page 5, Line 20) Page 5, line 13: I would recommend "Isotope analyses" or "Laboratory methods" instead of chemical analyses, because hydrochemistry is not discussed and isotope methods are no chemical methods. Reply: Accept. (see Page 5, Line 25) Page 5, line 17: ". . .Five groundwater samples . . ." In Table 3 seven ages are given for groundwater!? Reply: Sorry for the mistake. It should be 7 samples. (see Page 5, Line 29) Page 5, line 24. For the stable isotope analysis please give the specific gas preparation methods that were used, e.g., Gasbench, H-device, or TCEA? Reply: This information is presented in the revised manuscript. The gas preparation was performed with a Finnigan MAT Gas Bench. (see Page 6, Line 5)

Page 5, line 26. Please use appropriate definition of delta values. RSA/RST and not RVSMOW. This is especially important because you also give d13C values in Table 3. These are not defined against V-SMOW but VPDB I guess! Reply: The presentation is revised. The stable isotopic results are reported in the standard notation as $\delta$D, $\delta$18O and $\delta$13C values Eq. (1): $\delta$=(R_SA/R_STD -1)×1000‰ (1) where $\delta$ is the isotopic concentration of a sample, RSA is the isotope atom ratio D/H, 18O/16O or 13C/12C, RSTD is the corresponding isotope atom ratio of the international standard V-SMOW

for hydrogen and oxygen and VPDB for carbon. (see Page 6; Line 7-13)

Your precision is given as 1‰ and 0.1‰ for d2H and d18O respectively. Your d-excess results therefore should not be given with commas (see values in Tables as well). Reply: Accept.

What is the precision for your d13C values? What are the precision of your 3H values? Did you test any post corrections for 14C DIC? Reply: The analytical precision is $\pm0.3$‰ for $\delta$13C and ïĆś0.4 TU to ïĆś0.7 TU for 3H.The 14C dating correction method is described in Section 5.3. (see Page 11, Line 26-33; Page 12, Line 1-13)

Results and discussions Page 6, line 10-14: Please give d-excess values without digits. Reply: Accept.

Page 6, line 11: Why did you distinguish between groundwater and lake water? Please discuss results! Reply: That is because the difference in salinity between groundwater and lake water is significant (see the EC results in Table 2). The salinity of water would affect the stable isotope fractionation during evaporating. (see Page 6, Line 24, 25)

Page 6, line 22: You show that d-excess values are negatively correlated with d18O values (Figures 4b, 6b). Please discuss what this exactly means in your case. Usually these plots are used to argue for water vapor origin. Reply: Accept. (see Page 8, Line 8-11) Although the d-excess values are often used to infer atmospheric vapor sources, the evaporation experiments show that the d-excess values of water in the study area are primarily controlled by evaporation and decrease significantly but systematically with the extent of evaporation, providing another fingerprint for tracing the locally recharged water.

Page 6, line 27: "… as Fraction Modern (Fmdn)…" Usually given in percent modern carbon as pMC. See also Table 3. Reply: Accept. It is revised using pMC in the manuscript. (see Page 7, Line 12)

Page 8, line 10: "… from nearby IAEA GNIP ..." Please provide information on distance

and elevation of the station. Reply: Accept. (see Page 4, Line 28, 29)

You do not describe and discuss field parameters EC given in Table 2. Reply: The information of the EC given in Table 2 is presented in Section 4.1. (see Page 6, Line 24, 25)

The discussion on 14C free carbonate contribution to DIC is vague. Figure 9 is difficult to understand. Reply: This figure is not necessary after we made the correction of groundwater ages, so we delete the Figure 9. References: Please point out all Chinese references (in Chinese) for the international readers that do not understand Chinese language. Reply: Accept.

Figures and Tables: Figure 1. Please include location of GNIP station Zhanye in Fig. 1A) Reply: Accept. (see Page 18)

Figures 2 and 3 could be merged together Reply: Accept. (see Page 19)

Figure 6: The two diamond dots are not clearly visible, not visible in 6b. "Land water" should be rephrased! Soil water? Reply: Accept. We use two larger diamonds than the other ones to make the error bar of the dots more clearly. (see Page 22) It is explained in the figure caption. (see Page 22, Line 8-10) land water including groundwater (Li et al., 2016), rivers (average for each river) (Chen et al., 2012; Li et al., 2016), glacier snow melt water and frozen soil melt water (Li et al., 2016)

Figure 7: E – fluxes are misleading! Evaporation from groundwater to lake water fluxes could be better placed at boxes. Reply: Accept. (see Page 23) Table 1: d-excess values without digit. Reply: Accept. Table 2: Category should be rephrased into type. EC is given in mS not Ms! d18O and d2H measured against VSMOW. d-excess values without digit. Reply: Accept. Table 3: Temperature without digits or consistently. d13C against VPDB. Reply: Accept.

Technical corrections

Page 1, line 11: "(d2H-d18O) instead of (=d2H...) Reply: Accept. Page 1, line 25: "…

are scarce in arid regions, due ..." Please delete "in arid regions" Reply: Accept. Page 2, line 5: progress instead of progresses. Reply: Accept. Page 3, line 3:"... to104° ...." Space is missing. Reply: Accept. Page 5, line 20. Beta instead of Bata. Reply: Accept. Page 5, line 20. Beta Analytic Inc. (Miami, Florida, USA). Reply: Accept. Page 6, line 22: " ... and that it is negatively ..." instead of "...and strongly and negatively .." Reply: Accept.

Figures and Tables: Table 2. Table caption should include all columns. Location, date and EC is not mentioned. Reply: Accept. (see Page 26)

Table. 2 The EC, $\delta$18O and $\delta$D values of water samples from lakes, wells and spring in the BJD.

Figure caption Figure 5: delete space between d18 and O. Reply: Accept. Figure caption Figure 6: Please correct figure caption (e.g., d18O). Reply: Accept.
* * *
[Figure]

**Fig. 1.** Figure 1: Maps showing location of the Badain Jaran Desert (A), the Badain lake sampling area (B) and the Sumu Jaran lake sampling area (C). W34 in (A) is sampling site of Gates et al. (2008a).

[Figure]

**Fig. 2.** Figure 2: Schematic diagram showing the cross-section profile between the Sumu Jaran Lake and the Sumu Baran Jaran Lake as well as the water sampling points (A) and the groundwater flow direction (B)

[Figure]

**Fig. 3.** Figure 3: The relationship between $\delta$D and $\delta$18O (a) and between d-excess and $\delta$18O (b) of water samples from evaporation experiments.

[Figure]

**Fig. 4.** Figure 4: The $\delta$D vs. $\delta$18O plot of natural groundwater, lake water, and precipitation in the desert. Also shown are weighted monthly average and weighted annually average isotope ratios of precipitat

[Figure]

**Fig. 5.** Figure 5: The plot of $\delta$D vs $\delta$18O values (a) and d-excess vs $\delta$18O values (b) of groundwater and lake water samples from the BJD (red symbols), including new data from his study and previously published

[Figure]

**Fig. 6.** Figure 6: The conceptual model of the d-excess changing routines in the BJD. E represents evaporation.

[Figure]

**Fig. 7.** Figure 7: The flow model of groundwater near the Sumu Jaran Lake. The HCO3-concentrations of each wells are shown in the figure.

---

## Author Comment (AC2) · 27 May 2017

Reviewer 2ïïjŽ General opinion This manuscript (hess-2016-692; Water resources in the Badain Jaran Desert, China: New insight from isotopes Xiujie Wu, Xu-Sheng Wang, Yang Wang, and Bill X. Hu) is a concise presentation of the application of well-established isotope techniques in identification of groundwater sources for a group of desert lakes in China. A relatively small amount of isotopic data points to recent precipitation being the sole source of water in the coupled groundwater-surface water system. The isotopic evidence for that seems to be unequivocal, however, some questions (see specific comments) should be addressed in order to strengthen the line of reasoning. On the other hand, data on stable isotopic composition of DIC are not used at all and the part explaining 14C patterns requires some elaboration. Interpretation of presented data could be supported with a more detailed description of the hydrological

setting and limnology of the lakes. Text reads well, it is comprehensible and clearly presents the context, goals and conclusions of the study, however the description of the sampling campaign needs to be improved. My recommendation is to accept the manuscript after major revision.

Specific comments 1. Given that only few lakes were sampled a question arises, not addressed in the manuscript, if results of this study apply to all lakes of the Badain Jaran Desert. How big is the part of the desert where the lakes occur? It is not delineated on Fig. 1. Is the hydrogeological setting uniform throughout that part of the desert, so that the recharge – discharge pattern proposed in the manuscript might apply for the whole area? Reply: This is a constructive comment. Accordingly, in the revised manuscript, additional information and discussions are presented to address the question: (1) Section 2 has been significantly revised to show how the lakes are limited in the southeastern part of the desert and concentrated in a relatively small area (a 50-km-width rectangular zone around site-C in Fig. 1A) as well as how the aquifer system of sediments is relatively uniform in horizontal in the desert. At the regional scale, the Quaternary sediments are relatively uniform as fine sands and role as a continuous unconfined aquifer. The thickness of this aquifer is generally 100-300 m. (see Page 3, Line 7-9)

(2) We do not perform the discussion and analysis just based on our data that limited in a few lakes and sites. As shown in Figure 5, our data agree with the 2H-18O relationship revealed by most of the samples data in previous studies that covering most of the lakes and wells in the desert. In addition, the slope of the evaporation lines obtained in this study is comparable to the existing results (Chen et al., 2004; Chen et al., 2010; Wu et al., 2014).

(3) The limitation of the sample data is mentioned in the discussion part, Section 5.3. (see Page 12, Line 22-27)

2. Did conditions of evaporation during the evaporation experiments correspond to

the period of infiltration (also for recharge areas in the mountains)? Reply: In the revised version, this problem is analyzed and discussed in Section 5.2 as follow: (1) Our evaporation experiments were conducted in the summer and may not represent evaporation conditions in other seasons. However, the similarity between the evaporation line determined through our evaporation experiments (Fig. 3a) and those derived from measurements of natural water samples (Fig. 4) implies that seasonal variations in meteorological conditions do not significantly alter the evaporative $\delta$D-$\delta$18O pattern in the desert (see Page 8, Line 13-16) (2) Figure 5a also shows that the shallow groundwater samples from the Yabulai Mountain area fall above the evaporation lines in the BJD, but follow a trend line (EL Yabulai: y=4.2x $-$24.1) that extends through some of the lakes (Fig. 5a). This suggests that shallow groundwater in the Yabulai Mountain may be a source for some of the lakes in the BJD or the evaporation-infiltration conditions in the mountain areas were different from those in the desert. (see Page 10, Line 3-6) 3. (p. 2, line 25-27) Does this sentence refer to Qilian Mountains mentioned further in the discussion? Reply: The sentence is revised. It is the Qilian Mountains. (see Page 2; Line 25)

4. (p. 3, lines 8-9) What is the relevance of the information on the depth of root penetration? Reply: This sentence is misleading. It is removed in the revised version. 5. Chapter 3.1 Field Sampling. This part lacks some detail on the study area and sampling procedures. How many lakes were sampled? Five, or also some other of the "various" lakes? Provide their surface areas, maximum depths and volumes if known and explain if these characteristics are typical for the region. Provide information on wells depths and screening intervals as this is important in evaluation of well samples. Are the wells regularly used? If not, then 20 minute long pumping (with a peristaltic pump?) might not be enough to obtain a representative water sample from the aquifer. Reply: Accept. Additional descriptions are presented in Section 3.1. (see Page 4, Line 5-11) 6. (p6, lines 23-25) Was the good vertical mixing of lake waters confirmed by measurements of temperature and conductivity? Reply: Additional descriptions are presented in Section 4.2. Good mixture of water in the Sumu Barun Jaran Lake has

been also confirmed from the measurements of temperature and electric conductivity profiles in the summer (Chen et al., 2015). (see Page 7, Line 8-10) 7. (p. 9, lines 13-14) Are there any data on groundwater flow directions/hydraulic heads in the lake area to support this notion? Generally, the hydrogeological conditions are poorly explained in the manuscript. Reply: These sentences are misleading. They are deleted in the revised version. Section 2 has been significantly revised to express the hydrogeological conditions in the BJD. (see Page 3, Line 7-8, 16-23) 8. (p.10, lines 5-14) The bicarbonate and 13C DIC data do not support the gradual dilution of 14C with the old inorganic carbon derived from carbonate dissolution along the entire groundwater pathway. Both parameters change significantly only between points WS2 to WS1 pointing to isotopic exchange as a cause of the apparent increase of radiocarbon ages between WS4 to WS2. This discussion would benefit from any quantitative considerations on the isotopic evolution of DIC along the transect. However, the conditions of carbonate dissolution and isotopic exchange are controlled by aquifer properties which are not described in the manuscript. Reply: We agree that the varying 14C and 13C are not fully caused by the gradual dilution. In the revised manuscript, additional information about the aquifer properties are presented in Section 2 and Section 5.3. A quantitative analysis of the possible radiocarbon dilution factor is added and discussed in Section 5.3. The corrected 14C dating results are shown in Table 3. (see Page 3, Line 7-8, 16-23; see Page 11, Line 26-33; Page 12, Line 1-14) 9. (p.10, lines 27-29) The vertical homogeneity of lake water implies only good vertical mixing. The (isotopic?) steady state is a separate issue and can be assessed only from the temporal patterns. Is there any evidence for lake waters being in the steady state? By the way, what causes vertical mixing of the lakes? Wind or diurnal water temperature fluctuations? Reply: Accept! "Steady state" is not confirmed. In my opinion, the discharge of groundwater to the lakes is the most possible reason that courses vertical mixing. (see Page 12, Line 33) 10. Figs. 1B and 8 are not consistent with respect to the relative positions of wells and the Sumu Jaran Lake. The lake is located between W3 and W2 in Fig. 1B and to the west of W1 in Fig.8. Reply: The Fig 1B in last version is enlarged and changed to

Fig 1C and the exhibiting size of it is revised to clearly show the position of wells. (see Page 18) Technical comments 11. (p. 1, line 26) please remove "arid regions" Reply: It is revised in the manuscript. 12. (p. 7, line 22) "and ARE strongly" Reply: It is revised in the manuscript. 13. Fig. 7. Should be "brackish" not "blackish" Reply: It is revised in the manuscript.

[Figure]

**Fig. 1.** Figure 1: Maps showing location of the Badain Jaran Desert (A), the Badain lake sampling area (B) and the Sumu Jaran lake sampling area (C). W34 in (A) is sampling site of Gates et al. (2008a).

[Figure]

**Fig. 2.** Figure 2: Schematic diagram showing the cross-section profile between the Sumu Jaran Lake and the Sumu Baran Jaran Lake as well as the water sampling points (A) and the groundwater flow direction (B)

[Figure]

**Fig. 3.** Figure 7: The flow model of groundwater near the Sumu Jaran Lake. The HCO3-concentrations of each wells are shown in the figure.

---

## Author Comment (AC3) · 27 May 2017

The copy of a revised clean version of manuscript, the original manuscript with all the changes highlighted by using the track changes mode in MS Word, and the response writing in MS Word are attached in the supplementary documents.

Please also note the supplement to this comment:
http://www.hydrol-earth-syst-sci-discuss.net/hess-2016-692/hess-2016-692-AC3-supplement.zip

---

## Author Response (AR1)

Editor Decision: Reconsider after major revisions (further review by Editor and Referees) (12 Jun 2017) by Christine Stumpp

Comments to the Author:

At this stage of the review process, authors are only asked to answer to the reviewer comments and no revised manuscript is required yet. Therefore, my summary and decision is made solely on looking at the files containing the authors' response.

Two referees thoroughly evaluated the manuscript, and the authors answered in detail to the comments. The main points were:

1) more information about the methods is required

2) more information about the study site is required (lakes, hydrogeology)

3) improve data interpretation of 14C

4) consider conditions of evaporation during infiltration (experiment vs. recharge areas)

In addition, I want to emphasize that the authors still need to put the results into a broader context. My initial request before accepting the manuscript for HESSD was followed in the introduction. However, in the discussion and in the summary/conclusion results should also be put into a broader context too (e.g.: What has been found about water sources in other, similar desert regions? What can others learn that e.g. want to identify water sources in deserts at the other side of the world?).

These main and all other comments require some substantial rewriting of the manuscript according to the given answers. I ask the authors to carefully go through the text and -if not done yet- address my additional comment (broader context). A revised version of the manuscript can be uploaded and, I am happy to then reconsider the manuscript for publication after these major revisions were done.

Dear Editor,

Thank you so much for your helpful advice and suggestions! Further to our correspondence a couple of days ago, I'm attaching the revised version of my article entitled "Origin of water in the Badain Jaran Desert, China: New insight from isotopes". I have now completed all of the changes you requested.

We add some discussion to the **Discussion** and **Conclusion** parts.

**Response:**

**Discussion:**

**5.2**

Page 10, Line:19-23.

Similar results were found in arid central Australia (Tweed et al., 2011). Based on the stable isotope ($\delta^{18}O$ and $\delta^2H$) data, Tweed et al. (2011) suggested that groundwater was principally recharged during larger and intense rainfall events, but over longer timeframes, groundwater recharge was predominantly via diffuse processes rather than infiltration of floodwaters, even though the recharge may locally vary with distance from the floodplain.

And Page:10, Line: 32, 33 & Page:11, Line:1-8.

Only a few other studies also reported d-excess values and $\delta^{18}O$ values of groundwater from similar arid areas, such as Lake Eyre Basin (LEB), Australia (Tweed et al., 2011), and Jabal Hafit mountain in the United Arab Emirates (UAE) (Murad and Mirghni, 2012). Analysis of these previously published $d$-excess values and $\delta^{18}O$ values of groundwater from these arid areas also reveals strong relationships between the two (Fig. 7), suggesting similar recharge processes as observed in the BJD. This implies that previous interpretation in terms of the origin of groundwater may need to be revised. For example, Wood (2010) interpreted the negative $d$-excess ($-34.27‰ \sim -10.8‰$) values of paleo-groundwater as indicative of influx of evaporated runoff into the Red Sea during the last wet period resulting in the negative $d$-excess values in the moisture source. However, the strong relationship between the $d$-excess and $\delta^{18}O$ values of Gachsaran aquifer indicates that the water was affected by evaporation.

*Insert Figure 7*

Page: 24, Figure 7.

[Figure]

**Figure 7: Comparison of *d*-excess and δ¹⁸O values of groundwater samples from the BJD (red triangle), Lake Eyre Basin, Australia (green circle) (Tweed et al., 2011), Jabal Hafit mountain, UAE (dark green square) (Murad and Mirghni, 2012), and the two aquifers, Liwa and Gachsaran of Rub Al Khali, UAE (purple) (Wood, 2010). The trend lines are established and plotted in same color following the data.**

**Conclusion**

Page: 13, Line: 31

This study also demonstrated that the characteristic water isotopic patterns resulting from evaporation could be utilized to help resolve ambiguities in the interpretation of water isotope data in terms of recharge sources, especially, in the arid regions, such as the central Australia and the deserts of United Arab Emirates.